# A new parameterisation for homogeneous ice nucleation driven by highly variable dynamical forcings

Alena Kosareva[1], Stamen Dolaptchiev[1], Peter Spichtinger[2], and Ulrich Achatz[1]

[1]Institute for Atmospheric and Environmental Sciences, Goethe University Frankfurt, Frankfurt am Main, Germany
[2]Institute for Atmospheric Physics, Johannes Gutenberg University Mainz, Mainz, Germany

**Correspondence:** Alena Kosareva (kosareva@iau.uni-frankfurt.de)

**Abstract.** The present work aims to extend the parameterisation of homogeneous ice nucleation introduced in Dolaptchiev et al. (2023) by incorporating variable ice mean mass and generalizing the approach under different conditions. The proposed method involves introducing an empirically derived correction to the parameterisation based on a large data set of parcel model simulations to account for the mean ice particle mass variations. The method is validated against ensemble simulations using time-resolved double-moment ice microphysics, showing a mean relative error of less than 16%, with robust performance across a range of conditions. The uncertainty of the proposed parameterisation is evaluated for increasing integration time steps. The method remains computationally efficient and produces sufficiently accurate results, even with larger time steps, making it suitable for integration into numerical weather prediction and climate models. It is shown that the generalized approach not only provides a good representation of individual nucleation events but also effectively captures the statistics across the ensemble data. The prediction of the ice mixing ratio is also assessed against the reference double-moment system results. It is demonstrated that, following the nucleation event and re-initialization of the ice mixing ratio, the system equilibrates toward the reference solution within a few time steps. This refined parameterisation offers a more accurate prediction of ice number concentration and ice mixing ratio and is not limited to gravity wave-induced perturbations and can be supplemented by other relevant dynamical effects, such as large scale motions or even turbulence.

## 1 Introduction

Cirrus clouds correspond to up to 30% of the total cloud cover; however, they are still not well understood and their overall impact on climate in particular on the radiation forcing remains highly uncertain (e.g. Gasparini et al., 2018; Joos et al., 2014; Boucher et al., 2013). The albedo effect and the greenhouse effect of these clouds are similar in magnitude, meaning that the microphysical details, such as the number and shape of the ice crystals, can significantly influence their net radiative effect (Krämer et al., 2020; Matus and L'Ecuyer, 2017; Wang et al., 2020; Zhang et al., 1999). Some studies also indicate the indirect effects of cirrus clouds on future climate (e.g. Gettelman et al., 2012), which occurs mostly due to increases in ice crystal concentrations due to homogeneous nucleation from anthropogenic sulfur emissions.

These microphysical characteristics of cirrus are affected by a complex interaction of the small-scale cloud processes and the dynamics of the surrounding atmosphere. Observations suggest that gravity wave (GW) dynamics significantly impact the properties and life cycle of cirrus clouds (Kärcher and Ström, 2003; Atlas and Bretherton, 2023).

Vertical velocity perturbations, associated with gravity waves, turbulence, or other processes, lead to the homogeneous freezing of aqueous solution droplets (Koop et al., 2000; Baumgartner et al., 2022; Kärcher et al., 2024) at low temperature conditions in the tropopause region. The effect of gravity waves and other local dynamical processes on the nucleation has been widely investigated utilizing different approaches to describe the updrafts. Those interactions are studied, for example, in Dinh et al. (2016) and Kärcher and Podglajen (2019) utilizing a stochastic approach, in Joos et al. (2008) using a GW linear theory, and using the large-scale grid-resolved updraft velocity alongside subgrid-scale turbulence-induced updrafts in Zhou et al. (2016).

Based on the results from Baumgartner and Spichtinger (2019) studying the homogeneous nucleation due to constant updraft velocities, Dolaptchiev et al. (2023) has extended the asymptotic approach to incorporate GW dynamics. The present work complements Dolaptchiev et al. (2023) and has been designed for further usage in connection with GW parameterisations for a more realistic representation of GW-cirrus interactions in coarse-grid models. Recently, transient and lateral-propagation effects have been introduced into GW parameterisations (Kim et al., 2021; Achatz et al., 2023; Voelker et al., 2024), offering further potential for a more realistic simulation of these interactions.

The mean mass variability of the ice particles has a significant effect on the cloud ice number concentration prediction, which is also observed in numerical parcel model simulations in Dolaptchiev et al. (2023). The influence of a cloud ice particle size uncertainty in a climate model was assessed in Wang et al. (2020) and illustrates the sensitivity of the surface precipitation to the change in particle effective radius due to the larger energy imbalance, as well as spatial variability of short and long wave radiation. The effects of variable mass become more prominent in conditions of a higher number of pre-existing ice particles. Finally, it was previously shown (Gierens et al., 2003) that the deposition (or accommodation) coefficient, controlling the growth of ice crystals in the kinetic regime, largely affects the nucleation process, while at the same time being dependent on the radius or mass of ice crystals.

The current work focuses on generalizing the approach introduced in Dolaptchiev et al. (2023) to include mean mass variability for a wide range of initial conditions, e.g., dynamical forcing and environmental conditions. The asymptotic approach is supplemented by an empirical correction that can be used for various forcing representations. The proposed method is not restricted to the GW-induced dynamics and can be used for any kind of air motion, as e.g., large-scale updrafts or even turbulent fluctuations. This work is also relevant for a further implementation into a global numerical weather prediction (NWP) model, where it can be coupled to the dynamical parameterisations (e.g., GW and turbulence). However, due to the generalized formulation of the approach and robustness for larger time steps, it can also be applied to climate models with large time steps.

The article is organized as follows: Sect. 2.1 provides a brief overview of the approach used and the motivation for its generalization. Sections 2.2 and 2.3 describe the chosen approach for constructing the parameterisation and technical details of the used dataset. In the Sect. 3, the proposed parameterisation with correction is verified against reference calculations, and the range of applicability is tested based on ensemble calculations. Concluding remarks can be found in Sect. 4.

## 2 Theory and methodology

### 2.1 Double-moment approach to ice physics representation coupled to GW

In the present work, cirrus clouds are modeled using a double-moment bulk microphysics scheme (based on Spichtinger et al., 2023) that assumes an unimodal ice mass distribution function, where the zeroth and first moments of the distribution function are ice number concentration $n_\mathrm{i}$ and ice mixing ratio $q_\mathrm{i}$, respectively. Ice crystals are assumed to have a spherical shape, which simplifies the description of the properties. The mentioned assumption is reasonable for very small, newly nucleated ice particles formed through homogeneous nucleation. However, as ice particles grow through deposition, their shape becomes

more complex, introducing some uncertainty in the calculations. For consistency, we maintain the same shape assumption in deposition calculations, acknowledging that this introduces a certain error. Although a more sophisticated shape approximation would be beneficial for studying ice growth dynamics, this does not affect the developed correction and goes beyond the topic of the current study; former nucleation parameterisations also assume spherical ice particles at their early stage of life (see, e.g., Kärcher and Lohmann, 2002; Ren and Mackenzie, 2005; Barahona and Nenes, 2008). The ice physics scheme used here

is described in Spichtinger and Gierens (2009); Spreitzer et al. (2017) with the addition of sedimentation sinks. The influence of Lagrangian pressure and temperature variations, e.g., due to GWs, on homogeneous nucleation and as well as mean mass variability, are incorporated in Dolaptchiev et al. (2023). The resulting system of equations for $n_\mathrm{i}, q_\mathrm{i}$ and saturation ratio over ice $S$ reads:

$$\frac{Dn_\mathrm{i}}{Dt} = J\exp\left(B(S-S_\mathrm{c})\right) + \frac{1}{\rho}\frac{\partial(\rho n v_{n_\mathrm{i}})}{\partial z}, \tag{1}$$

$$\frac{Dq_\mathrm{i}}{Dt} = \frac{p_\mathrm{si}}{p}D\left(\frac{q_\mathrm{i}}{n_\mathrm{i}}\right)^{1/3}(S-1)Tn_\mathrm{i} + \hat{m}J\exp\left(B(S-S_\mathrm{c})\right) + \frac{1}{\rho}\frac{\partial(\rho q v_{q_\mathrm{i}})}{\partial z}, \tag{2}$$

$$\frac{DS}{Dt} = -D\left(\frac{q_\mathrm{i}}{n_\mathrm{i}}\right)^{1/3}(S-1)Tn_\mathrm{i} - \frac{p}{p_\mathrm{si}}J\exp\left(B(S-S_\mathrm{c})\right) - S\frac{1}{\pi}\frac{D\pi}{Dt}\left(\frac{L_\mathrm{i}}{TR_\mathrm{v}} - \frac{c_\mathrm{p}}{R}\right). \tag{3}$$

Here the saturation ratio over ice is described as $S = \frac{p_\mathrm{v}}{p_\mathrm{si}}$, $p_\mathrm{v}$ is the water vapor pressure, $p_\mathrm{si}$ is the saturation pressure over ice, $p$ is the pressure, $\hat{m}$ is the mean mass of newly nucleated ice particles, $J$ is a homogeneous nucleation rate, including also the number concentration of solution droplets, $B$ is a nucleation parameter, $S_\mathrm{c}$ is a critical saturation ratio, $D$ is a growth

coefficient, including the diffusion coefficient and the Howell factor, $v_{n_\mathrm{i}}, v_{q_\mathrm{i}}$ are sedimentation velocities, $T$ is the temperature, $\rho$ is the density, $\pi$ is the Exner pressure, $L_\mathrm{i}$ denotes the latent heat of sublimation, $c_\mathrm{p}$ is the specific heat capacity of air at constant pressure, $R_\mathrm{v}$ is the ideal gas constant for water vapor, and $R$ is the ideal gas constant for dry air, respectively. The values of the used parameters are presented in Table A1. In general formulation, the critical value of the saturation ratio over ice is temperature dependent; in the current work, the value is kept constant while other studies (such as Spichtinger et al. (2023))

show an insignificant influence of $S_\mathrm{c}$ on nucleation processes. Equations (1)-(3) are written in terms of material derivative $\frac{D}{Dt}$, i.e., all quantities are described following the motion of air parcels. The right-hand parts of the system of equations describe the ice crystal growth due to the deposition of water vapor, the formation of new ice crystals through homogeneous nucleation,

and the sedimentation of the ice crystals due to gravity. The latter is not included in the consideration further because of the much larger sedimentation time scale (see Dolaptchiev et al. (2023)).

In Baumgartner and Spichtinger (2019), detailed air parcel simulations are conducted using a box model and a bulk microphysics scheme (based on Spichtinger and Gierens, 2009), showing that for various environmental conditions, assuming a constant mean mass is generally valid over the duration of the nucleation. However, in case of pre-existing ice particles just before nucleation, when the saturation ratio exceeds one, the ice crystals' mean mass increases, leading to a higher deposition rate. This increased deposition affects the saturation ratio over ice (as a sink), which in turn influences the number of nucleated

ice crystals, reducing the actual nucleation rate. Given that it is crucial to include these effects in the current model, where the mean mass of ice crystals can be determined from the relation $q_i/n_i$. This is achieved by incorporating the evolution equation for $q_i$ and substituting $m(t) = q_i/n_i$ in the diffusional growth term.

Time scale separation is employed in Dolaptchiev et al. (2023) to assess the different processes and link the ice physics equations with the effects of GWs. The details of the coupling between the processes' time scales and assumptions of the GW

theory are presented in the Appendix A.

Further study is done considering the ice physics of a single air parcel influenced by GW dynamics, adopting a Lagrangian framework. In this case, material derivatives in the system of equations are replaced by time derivatives $\frac{D}{Dt} = \frac{d}{dt}$.

As shown in Dolaptchiev et al. (2023), considering only dominant processes, the system of Eqs. (1)-(3) can be reduced to the following form:

$$\frac{dn_i}{dt} = J \exp\left( B(S - S_c) \right), \tag{4}$$

$$\frac{dq_i}{dt} = \frac{p_{si}}{p} D \left( \frac{q_i}{n_i} \right)^{1/3} (S-1) T n_i, \tag{5}$$

$$\frac{dS}{dt} = -D \left( \frac{q_i}{n_i} \right)^{1/3} (S-1) T n_i + S F(t). \tag{6}$$

A forcing term $F(t)$ incorporating constant background updraft and GWs dynamics is written in the following form

$$F(t) = \frac{gL_i}{c_p R_v T^2} \left( w_{00} + \sum_j \hat{w}_j \cos(\omega_j t + \phi_j) \right),$$

where $g$ is the acceleration due to gravity, $w_{00}$ is a constant background updraft, $\hat{w}_j$, $\omega_j$, $\phi_j$ are the vertical wind amplitude, frequency, and phase of GW number $j$. It is worth mentioning that the application of the full system of Eqs. (1)-(3) is not a

standard approach, whereas the reduced system (4)-(6) is close to the reduced model in Baumgartner and Spichtinger (2019), and variations of it are commonly used in NWP and climate applications, however including the ice mass concentration.

Referring to the previous results from Dolaptchiev et al. (2023), in the case of constant mass, the asymptotic technique can be applied to obtain the prototype parameterization of the homogeneous nucleation process. By employing asymptotic expansions, the authors systematically decompose the governing equations into different scale regimes corresponding to the

key microphysical processes: nucleation and deposition. In the nucleation regime, where supersaturation rapidly increases due to gravity wave-induced cooling, they derive an analytical solution for the evolution of ice crystal number concentration.

In the growth (or deposition) regime, where ice crystals grow primarily by vapor deposition, the governing equations are solved separately under the assumption of gradual saturation adjustment. The solutions from both regimes are then matched using asymptotic matching techniques. This approach results in a reduced analytical expression for the number of nucleated ice crystals per nucleation event, explicitly linking it to wave parameters (such as frequency and amplitude) and background atmospheric conditions (such as temperature and humidity). The nucleation event is roughly determined where the saturation ratio reaches the critical value $S_c$, and the predicted value for the ice number concentration is found from:

$$n_{\text{post}} = \begin{cases} 2\dfrac{S_c F(t_0)}{D m_c^{1/3} T (S_c - 1)} - n_{\text{pre}}, & \text{if } n_{\text{pre}} < \dfrac{S_c F(t_0)}{D m_c^{1/3} T (S_c - 1)}; \\ n_{\text{pre}}, & \text{otherwise.} \end{cases} \tag{7}$$

Here $m_c = 10^{-12}$ kg is the reference mass the value, $n_{\text{pre}}$, $n_{\text{post}}$ are the ice number concentrations before and after the nucleation event, $t_0$ is the time of a nucleation event when $S = S_c$. Note that nucleation already occurs for values of $S$ close to $S_c$, see asymptotic analysis in Spichtinger et al. (2023).

In Dolaptchiev et al. (2023), the asymptotic approach was extended to account for the variable mass in the depositional growth coefficient in (5), (6) under the regime of slowly varying ice mixing ratio $q_i$ during the nucleation event. In such a case, the different processes can be decoupled, and only $n_i$ undergoes a quick change during the nucleation regime.

Numerical studies of the reduced system (4)-(6) based on a simplified parcel model showed several possible regimes relevant to the nucleation process (see Fig. 1). One regime at high values of the pre-existing ice crystal number concentration (shown in Fig. 1 in black) follows the same tendency as a constant mass case. The evolution of parameters can be separated into three phases (see Baumgartner and Spichtinger (2019)), where the nucleation part describes only the phase of rapid change of ice number concentration. In contrast, the saturation ratio and ice mixing ratio change slowly during this short time, and therefore can be assumed not to vary during this phase. However, conditions of low-to-zero pre-existing ice (shown in red) lead to a different behavior due to the growth coefficient change. Assumptions of the time separation for the nucleation regime do not hold, since the ice mixing ratio is changing on the same time scale as the ice number concentration. The extension to be proposed here aims to cover the whole range of initial conditions, allowing the use of the parameterisation accounting for mass variability independently from the regime.

## 2.2 Correction of the growth coefficient for variable mean mass

In the present study, we are utilizing the previously obtained result for the constant mass case (7). This approach would allow for a clear physical relation between $n_{\text{post}}$, the initial number concentration $n_{\text{init}}$, and the forcing term $F(t_0)$ at the point $t_0$ of the nucleation event. It would also account for different background conditions for temperature by design. Since mass variability changes the growth coefficient directly, as one can see from (6), the straightforward solution is to correct the growth coefficient in the constant mass parameterisation to a factor representing the mass at the point of the nucleation event. The corrected

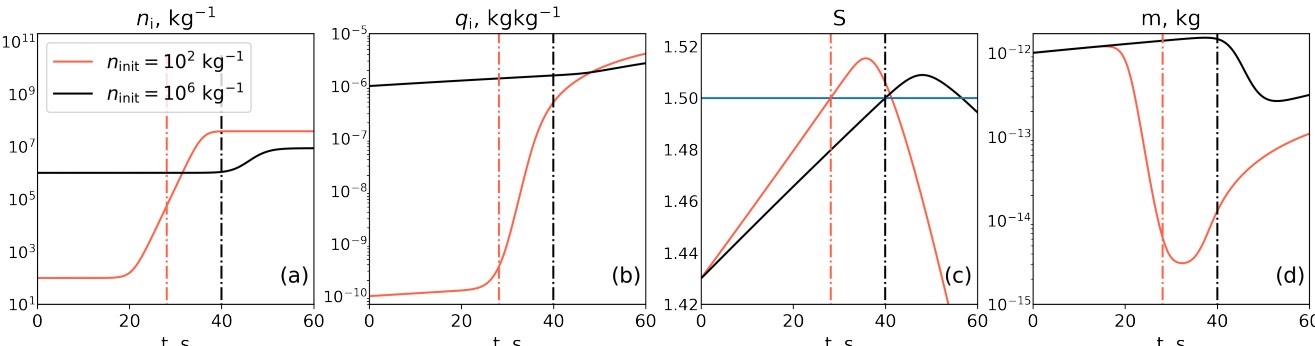

**Figure 1.** Evolution of the ice number concentration $n_i$, ice mixing ratio $q_i$, saturation ratio over ice $S$ and mean mass of ice particles $m = q_i/n_i$ calculated based on system (4)-(6). Two cases of initial conditions: low pre-existing ice $n_{init} = 10^2$ kg$^{-1}$ and higher values of pre-existing ice $n_{init} = 10^6$ kg$^{-1}$. The blue line corresponds to the critical saturation ratio, dashed lines mark the time of the nucleation event, where $S = S_c$ in red for the case $n_{init} = 10^2$ kg$^{-1}$, in black for the case $n_{init} = 10^6$ kg$^{-1}$.

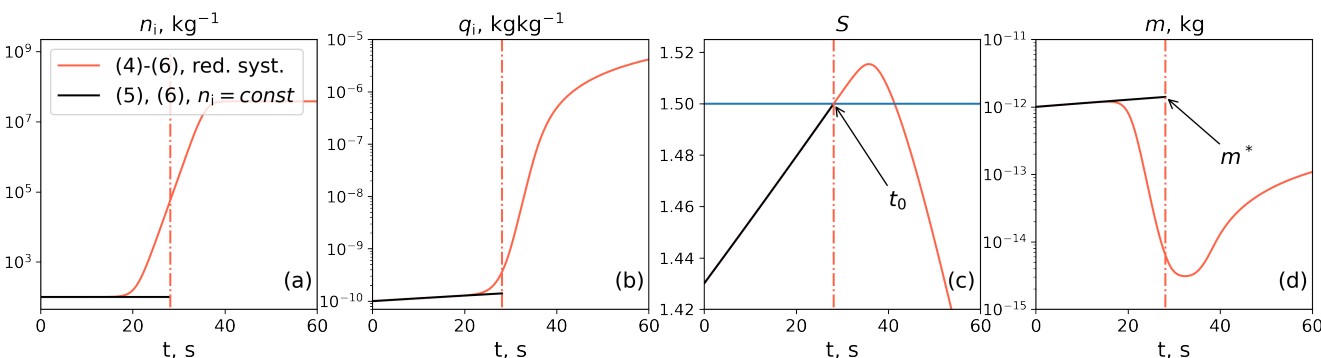

**Figure 2.** Evolution of $n_i, q_i, S, m$ in the reduced system (4)-(6) case and in the pre-nucleation regime, solving (5), (6) with assumption of $n_i = n_{init} = const$. The blue line corresponds to the critical saturation ratio, the dashed lines mark the time of the nucleation event, where $S = S_c$.

parameterisation has the following form:

$$n_{post} = \begin{cases} 2\dfrac{S_c F(t_0)}{D\left(m_{eff}\right)^{1/3} T(S_c-1)} - n_{pre}, & \text{if } n_{pre} < \dfrac{S_c F(t_0)}{D\left(m_{eff}\right)^{1/3} T(S_c-1)}; \\ n_{pre}, & \text{otherwise}, \end{cases} \tag{8}$$

where $\left(m_{eff}\right)^{1/3}$ is a growth coefficient correction factor representing the effective mean mass of the ice crystals at the start of the nucleation event.

150    In order to find the desired growth coefficient, one needs to define the parameters that can be used for the correction. Figure 2 shows the evolution of the parameters calculated from the reduced system (4)-(6) and known pre-nucleation regime parameters,

which can be used for deriving the correction and parameterisation. Here a pre-nucleation regime is defined as solution of the (4)-(6) system in the regime, where initial ice number concentration stays constant, which leads to solution of Eqs. (5),(6) with $n = n_{init} = const$ decoupled from Eq. 4. Sharp changes in the ice mixing ratio and mass in the case of a reduced system make it impossible to approximate the necessary quantities $q_i(t_0)$ or $\frac{q_i(t_0)}{n_i(t_0)}$ from pre-nucleation regime values itself. Therefore, the introduced effective mass is approximated based on a large data set. Variables that can be used in the approximation include the forcing term at the point of the nucleation event $F(t_0)$, the initial number concentration $n_{init}$, and the mass $m^*$ predicted from the pre-nucleation regime, assuming no change in ice number concentration before the nucleation event.

The exact effective mass $m_{eff}$ is determined from the results of the full system, such that the constant-mass approach leads to the observed $n_{post}$. Assuming a known evolution of the parameters during the nucleation event, Eq. (8), under the conditions allowing for further nucleation, can be solved for $m_{eff}$:

$$m_{eff} = \left( \frac{2 S_c F(t_0)}{(n_{post} + n_{pre}) DT (S_c - 1)} \right)^3, \tag{9}$$

where $t_0$, $n_{pre}$, $n_{post}$ are known from the resolved full system (1)-(3). A similar approach was used in Baumgartner and Spichtinger (2019), where the increase of the ice mass during the nucleation was found to be linear with a slope proportional to some constant mean mass.

Next, we discuss the construction of the data set from ensemble simulations based on Eqs. (1)-(3), disregarding the sedimentation process. This simplification can be justified by the small sizes of ice particles during and shortly after the nucleation event. The variability during the nucleation event depends on the initial conditions, such as ice number concentration, ice mixing ratio, or mean mass of ice particles, and on the GW forcing. Since we want to represent the nucleation process itself, the initial saturation ratio is set to a value of $S(t = 0) = 1.4$, close to the critical value. Other initial conditions are varied randomly using a uniform distribution in logarithmic space, i.e., $\log(n_{init})$ in physically meaningful ranges motivated by observations (Krämer et al., 2016): we choose $10^{-4} \mathrm{kg}^{-1} < n_{init}(t = 0) < 10^7 \ \mathrm{kg}^{-1}$, and $10^{-16} \ \mathrm{kg} < m(t = 0) < 10^{-12} \ \mathrm{kg}$. The lower bound for the initial number concentration is chosen to represent the conditions of the low-to-no pre-existing ice, which can occur in applications to the global NWP models.

Construction of the forcing is done based on the output from the global ICON (ICOsahedral Non-hydrostatic) model (Zängl et al., 2015) coupled with the GW parameterisation Multi-Scale Gravity Wave Model (MS-GWaM) (Bölöni et al., 2021; Kim et al., 2021; Achatz et al., 2023; Kim et al., 2023; Voelker et al., 2024). This parameterisation is based on Wentzel–Kramers–Brillouin (WKB) theory and is implemented using Lagrangian ray volumes, which are considered as carriers of the GW fields' wave-action density (Bölöni et al., 2021). The current version of MS-GWaM allows to account for convectively generated GWs and GWs generated from processes other than convection or flow over mountains. The influence of the GWs generated by the mountains is incorporated by using the original scheme (Lott and Miller, 1997) included in the ICON model. For constructing the forcing term, we utilize ICON version 2.6.5-nwp1, with 120 vertical levels and at horizontal resolution R2B5 ($\sim 80$ km). The initial conditions are taken for the 1st of May 2010, and the model is integrated forward for 3 weeks, accounting for 2 weeks of spin-up. Information on subgrid-scale gravity wave related vertical wind perturbations is retrieved from MS-GWaM using the ray volume with the maximum vertical wind amplitude per cell and the corresponding

frequency. In order to cover the variety of possible forcing term configurations, the following cases are included in the data set construction:

1. Large-scale vertical velocity from ICON run $w_{00}$, in such a case the forcing term reads: $F(t) = \frac{gL_i}{c_p R_v T^2} w_{00}$;

2. A single GW from MS-GWaM with maximum wind amplitude $\hat{w}$ within the cell, the corresponding frequency $\omega$ and initial phase $\phi$ which is picked randomly: $F(t) = \frac{gL_i \hat{w}}{c_p R_v T^2} \cos(\omega t + \phi)$;

3. A superposition of GWs, where the main GW is taken from the MS-GWaM and other GWs have randomly varying amplitudes $A_j$, frequencies $\omega_j$ and phases $\phi_j$ independent from the main GW and rescaled in a way, that the total momentum flux added is 1mPa, similar to approach used in Dolaptchiev et al. (2023): $F(t) = \frac{gL_i \hat{w}}{c_p R_v T^2} \cos(\omega t + \phi) + \sum_j \frac{gL_i A_j}{c_p R_v T^2} \cos(\omega_j t + \phi_j)$;

4. As in case 2 in the presence of (large scale) background updraft $w_{00}$:
$F(t) = \frac{gL_i}{c_p R_v T^2} \left( w_{00} + \hat{w} \cos(\omega t + \phi) \right)$;

5. As in case 3 in the presence of (large scale) background updraft $w_{00}$:
$F(t) = \frac{gL_i}{c_p R_v T^2} \left( w_{00} + \hat{w} \cos(\omega t + \phi) + \sum_j \frac{gL_i A_j}{c_p R_v T^2} \cos(\omega_j t + \phi_j) \right)$.

It is worth mentioning that the GW diagnosed from the MS-GWaM model dominates in cases 3 and 5 due to the larger amplitudes of the vertical velocity perturbations.

Distributions of the large-scale vertical winds and vertical winds from the GWs used for constructing the forcing are presented in Fig. 3 (a, b). The probabilities for the corresponding frequency $\omega$ are also presented, as well as those for the calculated GW momentum flux $\rho u w$ for the considered GW (see Fig. 3 (c, d)). The data is taken from the instantaneous output of the global ICON simulation at altitudes from 8 to 14 km, most relevant for cirrus formation. While this range does not explicitly cover the full tropical tropopause layer (TTL), including altitudes up to 18 km, we note that the selected range captures the majority of ice formation conditions observed in the Upper Troposphere/Lower Stratosphere (UTLS). Moreover, the vertical velocity distributions in this range are comparable to those at higher altitudes, and the proposed parameterization is additionally tested under conditions representative of stronger forcing (see Appendix C). These tests suggest that the approach remains robust and physically consistent even beyond the original range of sampled data.

It is seen in Fig. 3 that the large-scale updraft varies in range from -0.2 ms$^{-1}$ up to 0.4 ms$^{-1}$, where in the majority of the cells, vertical velocity is close to zero. Therefore, we include cases 2) and 3) without the large-scale updraft when creating the dataset. Vertical wind perturbations $\hat{w}$ obtained from the GW parameterisation show stronger vertical velocity, with a rare maximum above 2 ms$^{-1}$. Corresponding frequencies $\omega$ are sampled in a range between Coriolis parameter $f = 10^{-4}$ s$^{-1}$ and buoyancy frequency $N = 2 \cdot 10^{-2}$ s$^{-1}$ in the tropopause region and the form of the $\omega$ spectrum resembles the Desaubies form (VanZandt, 1982). It is worth noting that the resulting momentum flux based on the highest-amplitude ray volume is aligned with the observational statistics obtained from the balloon measurements (Hertzog et al., 2012) and previously calculated momentum fluxes from the MS-GWaM parameterisation (Kim et al., 2021). Therefore, the total momentum flux considered

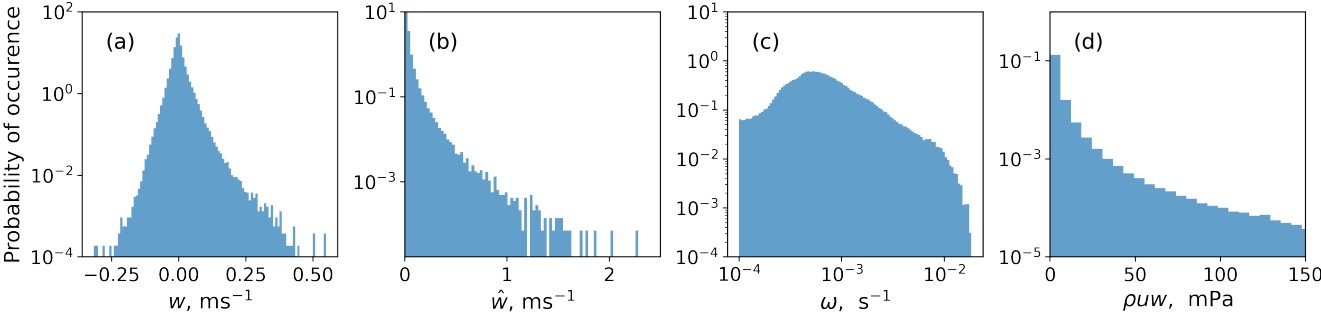

**Figure 3.** Frequencies of occurrences of (a) the large-scale vertical velocity $w$, (b) the vertical velocity amplitude $\hat{w}$, (c) the frequency of the GWs $\omega$ defined from the MS-GWaM parameterisation, and (d) the momentum flux corresponding to the detected GW.

lies in the physical range, and the forcing term applied in the parcel simulations is realistic to some degree. Further support for the realism of the applied conditions comes from Fig. 3 b, where the PDF of gravity wave–induced vertical velocity perturbations $\hat{w}$. The spread and frequency of occurrence show a good agreement with available observations from balloon measurements (Podglajen et al., 2016; Köhler et al., 2023).

The variation of vertical velocities used in the construction of the forcing is sampled from the wide range of conditions, including the large updrafts up to $10 \ \mathrm{ms}^{-1}$. Because of the general model formulation, one can add the vertical velocity contributions to the forcing term based on other information provided by the model, for example, by the turbulence scheme. However, such an extension of the forcings is not tested in the current work.

The exact value of the effective mass $m_{\mathrm{eff}}$ is found from Eq. (9) based on the ensemble calculations for $\sim 2 \cdot 10^5$ nucleation events; the results are shown in Fig. 4. The data set is created using random initial conditions for $n_{\mathrm{i}}$, $q_{\mathrm{i}}$, and various GW forcing configurations as discussed above. The internal variability of the data is quite high, especially depending on the forcing term at the point of the nucleation event. However, by fixing certain parameters (blue and black markers), one can see a clear dependence of $m_{\mathrm{eff}}$ on the chosen parameters. The variability with the forcing term is much larger, whereas under the conditions of smaller initial number concentrations, $m_{\mathrm{eff}}$ stays nearly constant. The dependence on the forcing term is decaying with large variability for smaller values of the forcing at the point of the nucleation event. The increase of the mass with an increase of the initial number concentration is connected to the fact that pre-existing ice suppresses the nucleation process, hence, the number of newly nucleated ice crystals is smaller. Because the initial ice mean mass is quite high, and the new particles are four orders of magnitude lighter, the recalculated mean mass would increase if more pre-existing ice is present. The observed behavior of the mass in dependence on the forcing term is the opposite, it decreases with increasing $F(t_0)$. A larger forcing term at the point of the nucleation is associated with the larger updrafts leading to a faster change in the saturation ratio over ice. The depletion of the water vapor via deposition in such conditions is ineffective, leading to smaller mass values.

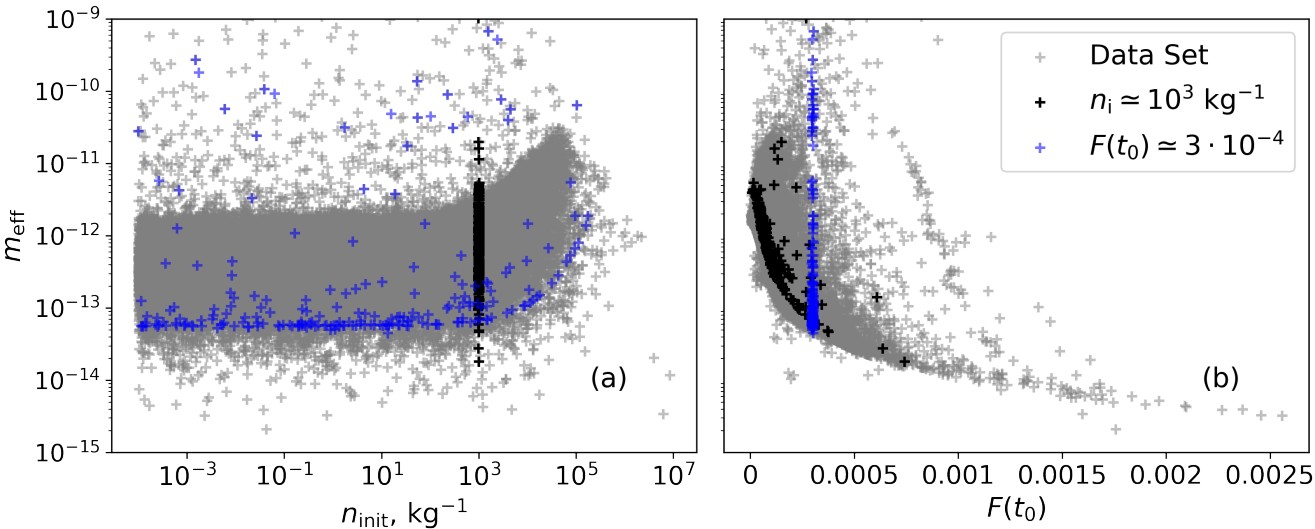

**Figure 4.** The exact $m_{\text{eff}}$ correction depending on the initial number concentration (a) and on the forcing term value at the nucleation event point (b). Blue markers correspond to calculations with constant $n_{\text{init}}$, black markers correspond to calculations made with approximately constant values of $F(t_0)$.

### 2.3 Mean mass fitting for parameterisation

240  In the present section, we consider options for approximating the dependence of $m_{\text{eff}}$ based on the parameters known from the pre-nucleation regime. Since the evolution of the ice mixing ratio $q_i$ is not captured in this regime (due to fixed growth coefficient and $n_i$), we propose to fit a function depending on the known parameters, such as initial ice number concentration $n_{\text{init}}$ and forcing term at the point of the nucleation event $F(t_0)$, to the ensemble simulations results. These parameters are selected based on the definition of the correction given in Eq. (9), and their relevance is further supported by the clear dependence

245  observed in Fig. 4, where only a slight spread is visible in cases with fixed $n_{\text{init}}$ and $F(t_0)$.

Based on the created data set and the relative behavior of $m_{\text{eff}}$ in dependence on $n_{\text{init}}$ and $F(t_0)$, an approximation to the data is derived. The non-linear least squares method is applied to fit an assumed function to the known data from the ensemble simulations. The method requires to suggest a certain functional dependency on the parameters. Given the data set behavior, the chosen simplified function is:

250
$$m_{\text{eff}}(F(t_0), n_{\text{init}}) = \exp(a_1 + a_2 F(t_0)^{1/3} + a_3 n_{\text{init}}^{1/3} + a_4 n_{\text{init}} F(t_0)^{1/3}). \tag{10}$$

The mean error for $m_{\text{eff}}$ during the target data set reproduction is around 30%, which translates into a mean error of $\sim 10\%$ in radius, for the set of $\sim 2 \cdot 10^5$ nucleation events with different GW forcing representations.

As an extension, a more complex dependency on $n_{\text{init}}$, $F(t_0)$, and involving one more parameter $m^*$, defined from the pre-nucleation regime, is introduced and tested. The results showed an improvement compared to the previously introduced fit (10);

255  see the corresponding figures and discussion in the Appendix B. While the use of an advanced fit improves the prediction of the

effective mass ($m_{\text{eff}}$), the ice number concentration after nucleation ($n_{\text{post}}$), which is the primary target of the parameterization, is already well captured by the simple fit. The more complex approach yields only a marginal improvement of about 2% in the mean error. Besides, the advanced fit adds more complexity to the calculation, requires the use of an additional parameter and because of the additional terms used can lead to diverging results in cases out of sample range with regard to the reference data used for the parameter optimization.Therefore, given the efficiency and robustness of the original simple fit, we recommend its use and present all further analysis based on this version.

## 3   Results. Evaluation of the Model

Further results are shown for an independent data set consisting of 490 nucleation events in order to validate the chosen fit. Initial conditions for $n_{\text{i}}$ and $m_{\text{i}}$ were created randomly, and the forcing term is based on ICON simulation is different from the one used for the construction of the fit. In the following, we will first asses the proposed fit. This will be followed by stability and accuracy tests, depending on the time step chosen for the integration of the Eqs. (5)-(6) in the parameterised case.

We acknowledge that developing a parameterization based on a limited dataset is inherently challenging and introduces certain limitations. The considered conditions are representative of the UTLS region and are specific to the processes as captured by the ICON model. As a result, extreme or anomalous cases may be underrepresented. Nevertheless, we argue that the physically grounded formulation of the proposed parameterization, along with its robustness to variations in forcing, allows it to provide reasonable predictions even outside the original fitting range. To support this claim, we validated the parameterization on an independent dataset and included results for selected test cases with unusually strong vertical velocities in the Appendix C.

### 3.1   Extended parameterisation

A comparison of the application of the chosen fit (10) and the exact effective mass calculated from the nucleation event simulations is shown in Fig. 5. A direct comparison of predictions with the exact values for $m_{\text{eff}}$ shows a good agreement for the major part of the data points. Nonetheless, the fit is not able to capture the higher possible $m_{\text{eff}}$ values at higher values of $n_{\text{init}}$. Fig. 6 shows the exact post-nucleation ice number concentration and the one obtained with the use of the extended parameterisation (8). The underestimation of the $m_{\text{eff}}$ values at higher values of pre-existing ice number concentrations leads to an overestimation of the ice number concentration as shown in Fig. 6.

Referring to the Fig. 5, the most frequently occurring cases of $m_{\text{eff}}$ in the range between $10^{-13}$ kg and $10^{-12}$ kg, and the corresponding final $n_{\text{post}}$ in Fig. 6, in the range between $10^5$ and $10^6$, are successfully represented by the chosen fit (10) and the corrected parameterisation (8). However, the outlying extreme cases are poorly captured due to the simplicity of the fit. The overall behavior of the fit stays similar to the exact data; therefore, with decreasing $n_{\text{init}}$, the prediction for the $m_{\text{eff}}$ stays nearly independent from $n_{\text{init}}$ (see Fig. 5 (a). The cases with an increased forcing term correspond to the smaller values of $m_{\text{eff}}$, increasing the final prediction for $n_{\text{post}}$ in the exact data, and the same behavior is captured by the fit.

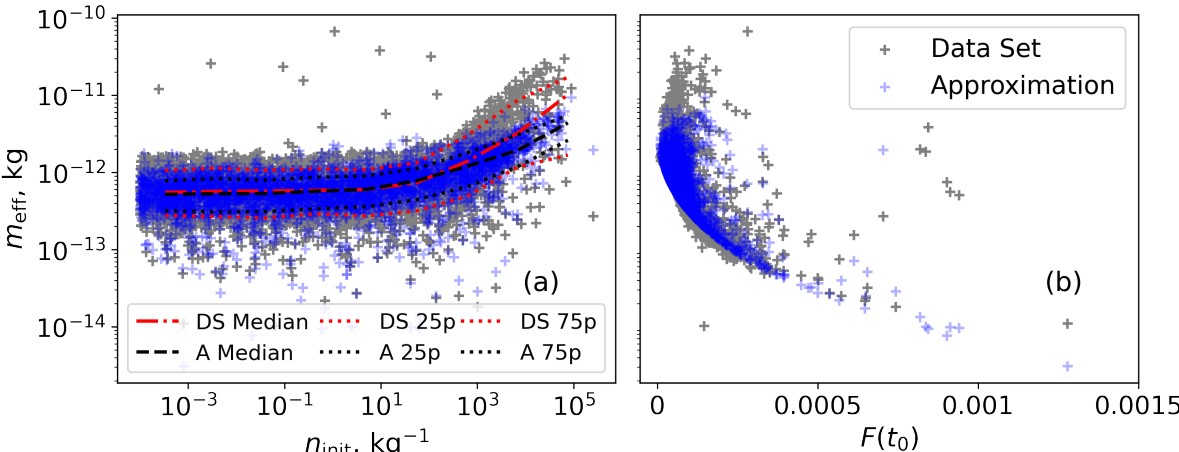

**Figure 5.** Exact $m_{\text{eff}}$ correction depending on the initial number concentration and $m_{\text{eff}}$ prediction from the constructed fit (10) for independent ensemble calculations.

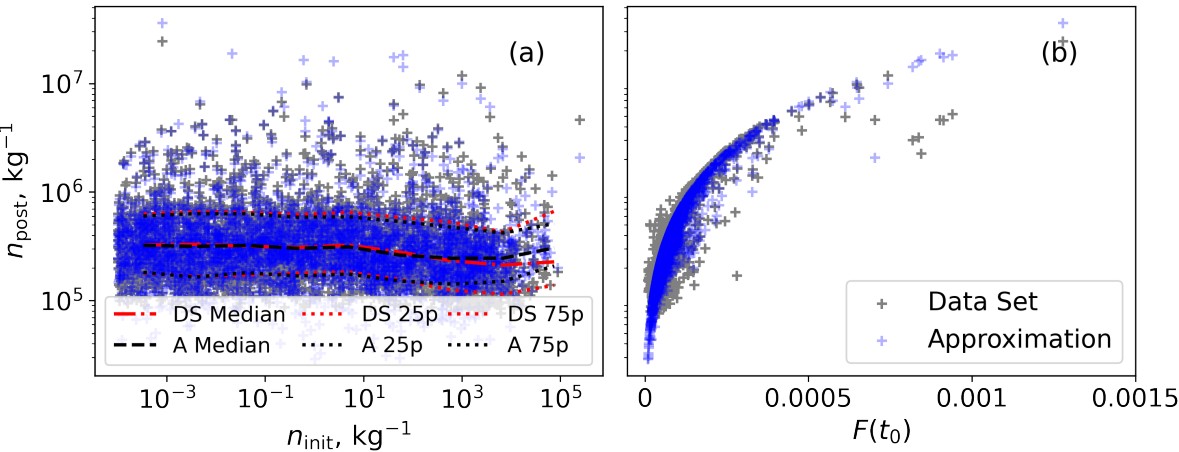

**Figure 6.** Post-nucleation ice number concentrations defined from the full system and prediction from the constructed parameterisation with correction (10).

To assess how well the chosen approach represents individual cases, the relative errors of $m_{\text{eff}}$, effective ice particle radius $R_{\text{eff}}$ and $n_{\text{post}}$ with respect to their exact values are calculated. The effective radius is calculated based on the effective mass as follows: $R_{\text{eff}} = \left(\frac{3 m_{\text{eff}}}{4 \pi \rho_{\text{i}}}\right)^{\frac{1}{3}}$. The probability density function (PDF) of the relative errors is presented in Fig. 7. Here the relative errors are calculated as $\Delta M = (M_{\text{predicted}} - M_{\text{exact}})/M_{\text{exact}} \cdot 100\,\%$, with $M = m_{\text{eff}}, R_{\text{eff}}, n_{\text{post}}$. The mean errors for $m_{\text{eff}}$ and $n_{\text{post}}$ predictions are 24% and 16%, respectively. Despite the substantial deviation in effective mass, the prediction for the effective radius shows a good agreement and the mean error is only about 10%. More importantly, 90% of the error for the prediction of

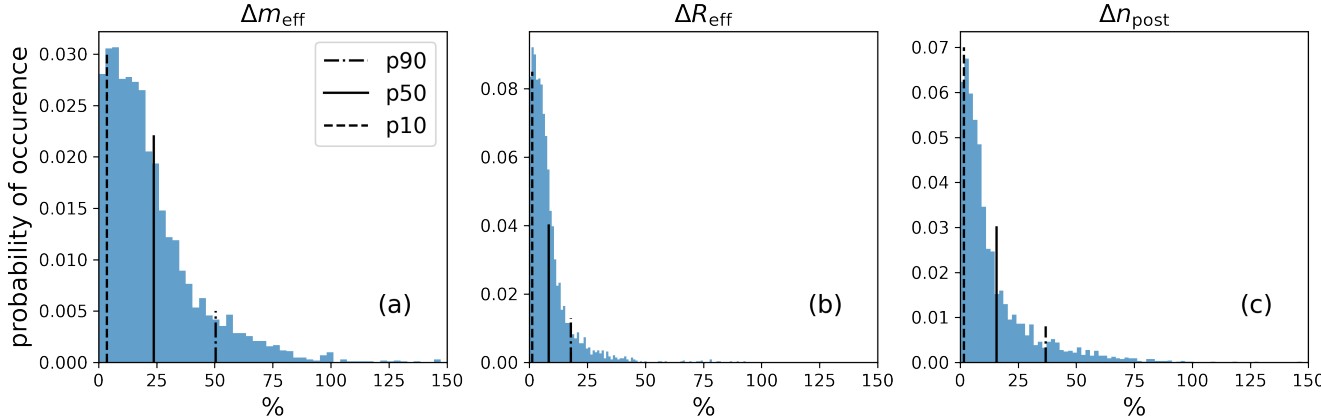

**Figure 7.** Probability density functions of the relative errors in the prediction of $m_{eff}$, $R_{eff}$ and $n_{post}$ with respect to the exact solutions. Vertical lines correspond to the 90th, 50th, and 10th percentiles of the distribution.

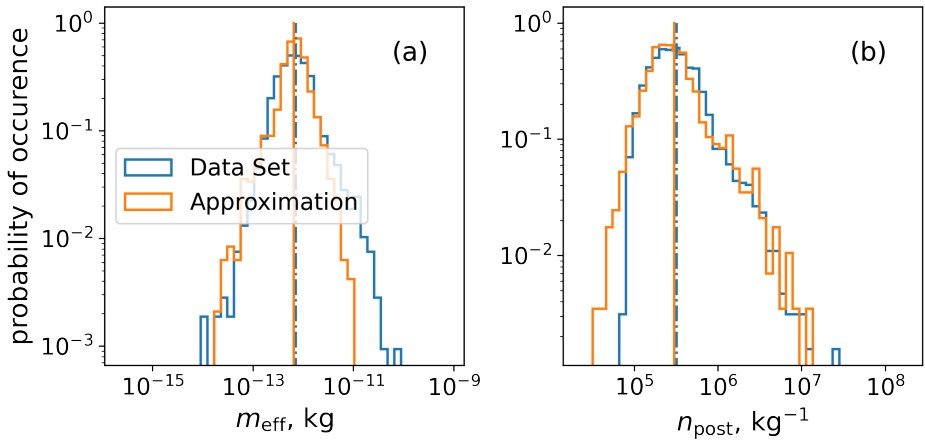

**Figure 8.** Probability density functions for $m_{eff}$ and $n_{post}$ exact ensemble solutions and predictions. Vertical lines correspond to the mean of the distribution.

the ice number concentration lies below 37%. The majority of the individual cases are represented sufficiently well, whereas outliers and poorly captured events with a deviation in $n_{post}$ above 50% correspond to less than 10% of the distribution.

For a better understanding of the prediction deviations, we also consider the PDF of the exact parameters and the approximated ones in order to assess the ability of our fit to capture the statistical properties of the dataset. Previous ensemble simulations using constant mass parameterisation by Dolaptchiev et al. (2023) showed that while individual events can vary significantly, or even be missed, the statistical properties for the ensemble still align well.

Comparisons of the PDFs for the effective mass $m_{eff}$ and the predicted ice number concentration are shown in Fig. 8.

The PDF of the predicted mass from Fig. 8 at the point of the nucleation exhibits a slight underestimation of the mean value, due to the inability to correctly predict larger $m_{\text{eff}}$ (also shown in Fig. 5). For the same reason, the tail of the distribution is shorter, and the maximum value has a mismatch of approximately one order of magnitude. The lower values of $m_{\text{eff}}$ are represented better, and the minimum values are in the same range. A comparison of the PDFs of the ice number concentration shows a good correspondence of major statistical indicators, such as mean, maximum, and minimum values. However, the extended parameterisation gives occasional overestimation of $n_{\text{post}}$ values for larger values of the forcing term and underestimation of the $n_{\text{post}}$ in cases of small forcing (see the right panel in Fig. 6).

Apart from the values of $n_{\text{i}}$, the prediction for the ice mixing ratio $q_{\text{i}}$ is made based on the estimated mass $m_{\text{eff}}$ and the ice number concentration $n_{\text{post}}$, calculated using the corrected parameterisation, via $q_{\text{i, post}} = m_{\text{eff}}\, n_{\text{post}}$. Such an approach contains some physical inconsistencies because the approximated ice mean mass $m_{\text{eff}}$ differs from the mean mass after the nucleation event, which is supposed to be used. However, the estimation gives a suitable approximation for the further simulation of $q_{\text{i}}$. Further, the evolution of the parameters for two isolated cases is presented in Figs. 13 and 14 to show the $q_{\text{i}}$ prediction relevance.

## 3.2  Influence of the time step

The results shown in Sect. 3.1 are conducted using the same time step for the solution of the full system (1)-(3) and coupled system (5),(6) employed in the parameterisation. Such a decision is made to keep the results comparable without introducing further uncertainties. However, the major goal of the parameterisation is to increase the numerical performance, especially by using larger time steps as in coarse resolution models, e.g., NWP and climate models. Therefore, the following section is devoted to the depiction of the uncertainties and evaluation of the results under different time steps.

The investigation of the stability of numerical integration and applicability of the defined approach is done based on a data set with 6 GWs and a constant background updraft $w_{00}$. One of the sources of uncertainty is the accuracy of the detection of the forcing term at the time of the nucleation where $S = S_{\text{c}}$. The nucleation event can be misplaced, changing the defined time of the nucleation event $t_0$, depending on the time step used for the calculation of the saturation ratio evolution (6). The calculation of the forcing term is sensitive to the changes in $t_0$, which leads to further errors in the estimation of the $F(t_0)$ and other parameters.

Further, the relative errors of the predictions from the exact values $\Delta M$ are calculated as mentioned before in Sect. 3.1 for the considered parameters $M = F(t_0), m_{\text{eff}}, n_{\text{post}}$. First, in Fig. 9 we compare relative errors in the forcing $F(t_0)$ when solving slow-scale processes for $S, q_{\text{i}}$ based on system (5) and (6) with five different time steps choices. In Figs. 9-11, horizontal bars represent percentiles of the distribution to illustrate data variability. Note that in several cases, the 10th percentile lies near zero and may not be easily distinguishable in the plots due to scale limitations. The smallest chosen time step of one second gives the exact solution, whereas longer time steps lead to deviations from the exact prediction. Such deviations may influence the estimated $m_{\text{eff}}$ via errors introduced in $t_0$ and $F(t_0)$ determination. However, the crucial impact it would have on the $n_{\text{post}}$ prediction by directly changing $F(t_0)$ in the formula (8). Maximum differences above 50% occur rarely and are associated with outliers, for instance, some cases where the nucleation event is missed with the usage of the larger time step. One such

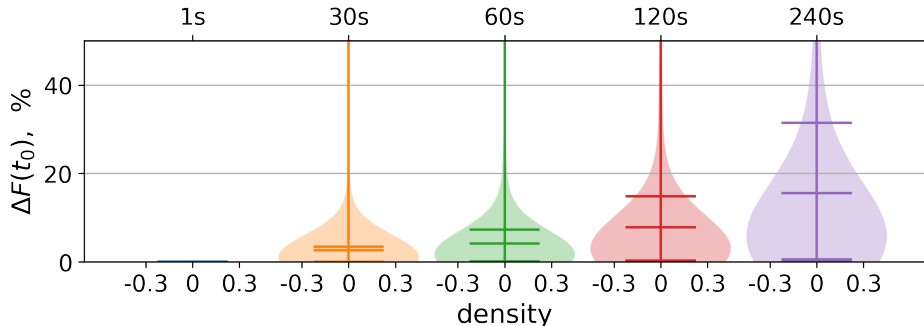

**Figure 9.** Probability density function of the relative error of the parameterized forcing term $F(t_0)$ compared to the value detected from the exact full-system solution in %. Horizontal bars correspond to the 90, 50, and 10 percentiles of the distribution.

case is illustrated further in this section in Fig. 13. The mean relative error in the forcing term corresponding to the case of 30

second time steps are 2.6%, and the 90th percentile is 3.5%. The mean and the 90th percentile of the PDF are doubled with doubling the time step and rise up to 15% and 31% for the largest considered time step of two minutes. Considering the exact differences instead of relative errors, it is found that larger time steps mostly lead to an underestimation of the forcing. Hence, because of the direct relation to the ice number concentration, one would expect underestimation of the $n_{post}$ with increasing time step.

Uncertainties from the detection of the forcing term then lead to a variation in $m_{eff}$, considered in Fig. 10. The overall error we introduce by using the fit is illustrated by the first PDF when the same time step as in the full system is used. The additional deviation introduced by differences in $F(t_0)$ leads to a variation of about 2-3% in the mean error of $m_{eff}$, which is negligible compared to the overall error introduced by the fit.

    Comparing the error of the ice number concentration prediction from the exact solution in Fig. 11, one observes the increase

in the errors when going to larger time steps. The mean and 90th percentile of the distribution stay nearly the same when

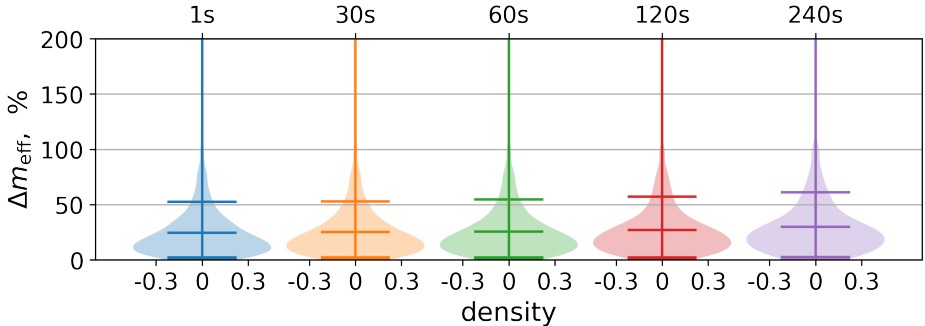

**Figure 10.** Probability density function of the relative error of $m_{eff}$ predicted by fit (10) with respect to the exact mass calculated from the full system using (9) in %. Horizontal bars correspond to the 90, 50, and 10 percentiles of the distribution.

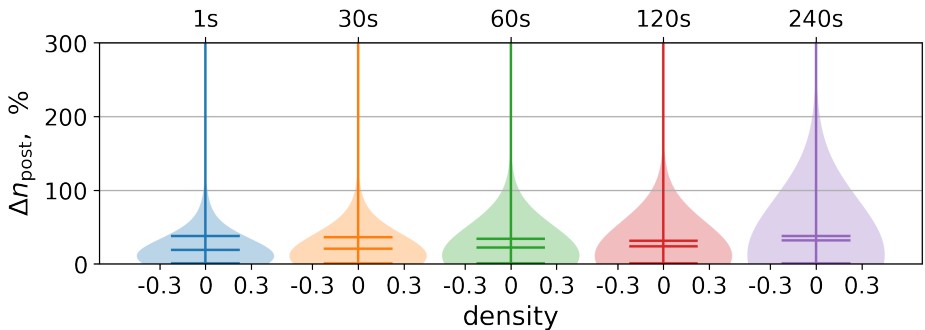

**Figure 11.** Probability density function of the relative error of $n_{post}$ predicted using fit (10) by (8) compared to the exact mass calculated from the full system in %. Horizontal bars correspond to the 90, 50, and 10 percentiles of the distribution.

increasing the time step from 1 second to 60 seconds. The increase in mean error becomes larger for a time step $dt = 120$ s, and at a time step $dt = 240$ s it becomes closer to 50%. We conclude that one can use a time step of 60 seconds without a significant error increase. To estimate the quality of the new parameterisation, one can compare the mean errors to the relative accuracy of measurements (see, e.g., Krämer et al., 2016, 2020). The resulting mean error using a time step of 60 seconds is

of the same order as the accuracy of measurements, which is about 15% (c.f. Luebke et al., 2016). Therefore, the proposed approach and chosen time step are well-suited for the representation of the process.

In order to check the dependence of the statistics on the time step used in integrating equations (5),(6), the comparison of the PDFs of $m_{eff}$ and $n_{post}$ is shown in Fig. 12. The overall shape of the approximated PDF of mass $m_{eff}$ agrees with the exact solution. However, the case with the largest time step of two minutes shows an underestimated $m_{eff}$ minimum for

several orders of magnitude. This leads to poor representation of the cases with higher $n_{post}$ and consequent overestimation of the $n_{post}$ maximum (see Fig. 12 lower panel). Predictions of the ice number concentrations are not far from the exact full system results if one compares the center of the PDF. The peak of the predicted $n_{post}$ PDFs stays in the range of 2% difference compared to the full system for all the parameterisation cases. However, the prediction made using the largest time step substantially overestimates the maximum $n_{post}$ value. It is worth mentioning that the ice number concentrations obtained

with the parameterisation and full system are in the range of the observed quantities for the homogeneous nucleation (Krämer et al., 2020).

Further, we present two nucleation events cases in order to directly compare the evolution and parameterisation prediction with different time steps (see Figs. 13 and 14). By means of such comparison, one can make an additional conclusion on the correctness and possibility of use of $q_i$ as mentioned earlier and assess numerical stability of the method under different time

steps. Initial conditions for the presented cases are picked from the ensemble calculation with 6 superposed GWs with the presence of a background updraft.

Both cases do not show any difference between the reduced system (4)-(6) and the full system (1)-(3). The corrected parameterisation (8) is tested for larger time steps of 30, 120, 240 s, omitting the case with the smallest time step. The first case

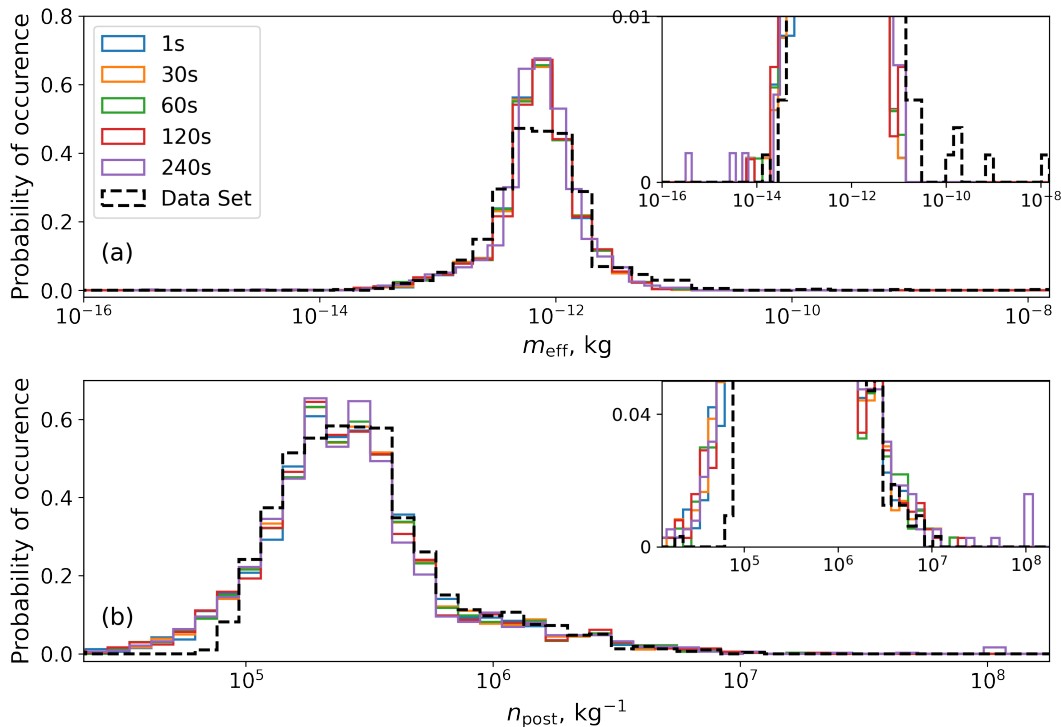

**Figure 12.** Probability density functions of $m_{eff}, n_{post}$ and $q_i$ from the full model and predicted by the parameterisation (8) with different used time steps.

in Fig. 13 illustrates that the parameterization gives a good agreement with the full system, even with larger time steps. The evolution of the saturation ratio over ice, ice mixing ratio, and mass is aligned with the full system results after some time following the nucleation event. The prediction of the ice number concentration is not that exact for cases with larger time steps, but the difference is still in the range of 30%. After the prediction of $q_i$ using the $m_{eff}$ and $n_{post}$ from the parameterisation one can see that integration is done without numerical problems. The evolution of the ice mixing ratio $q_i$ shows a jump due to the nucleation process, followed by a short relaxation period, after which the $q_i$ prediction is adjusted and overlaps with the full system result.

The second test case presented in Fig. 14 has been picked on purpose to show several possible outcomes of using the parameterisation and the consequences of usage of larger time steps. The evolution of the slowly-varying parameters such as $S, q_i, m$ is captured relatively well by the parameterisation with the time steps of 30, 120 seconds. The largest considered time step, however, leads to smaller variations in the saturation ratio and hence the nucleation event is not captured by the parameterisation.

The parameterisation using a time step of 30 seconds gives a sufficiently good agreement with a relative error compared to the exact solution not larger than 22%. However, even though the parameterisation with a 2-minute time step predicts the

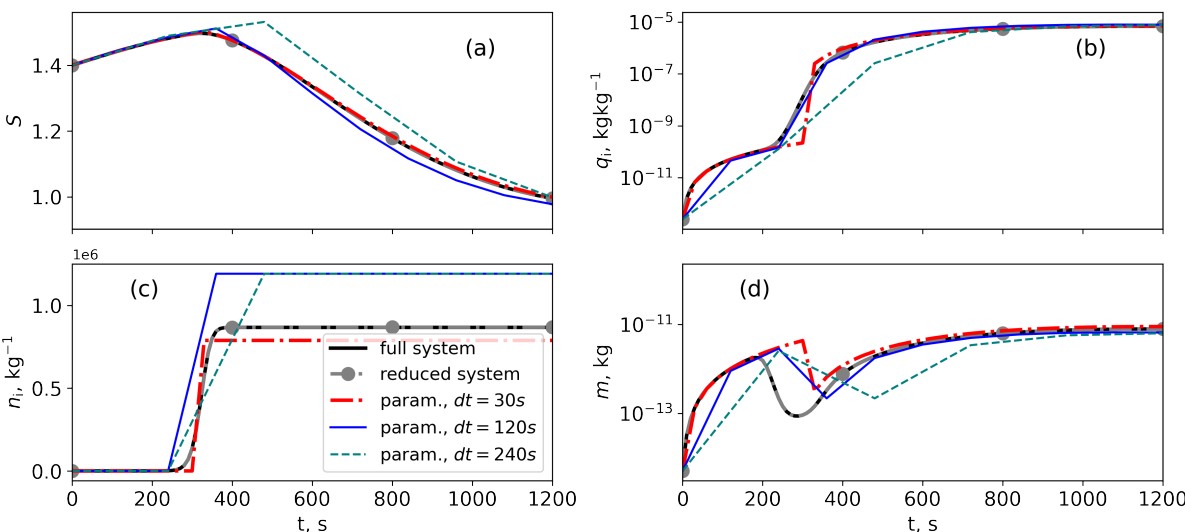

**Figure 13.** Comparison of the parameters evolution in full system, reduced system, and parameterisation with proposed correction. Initial conditions: $S = 1.4$, $n_\mathrm{i} = 50 \ \mathrm{kg}^{-1}$, $q_\mathrm{i} = 2.6 \cdot 10^{-13} \ \mathrm{kg\,kg}^{-1}$.

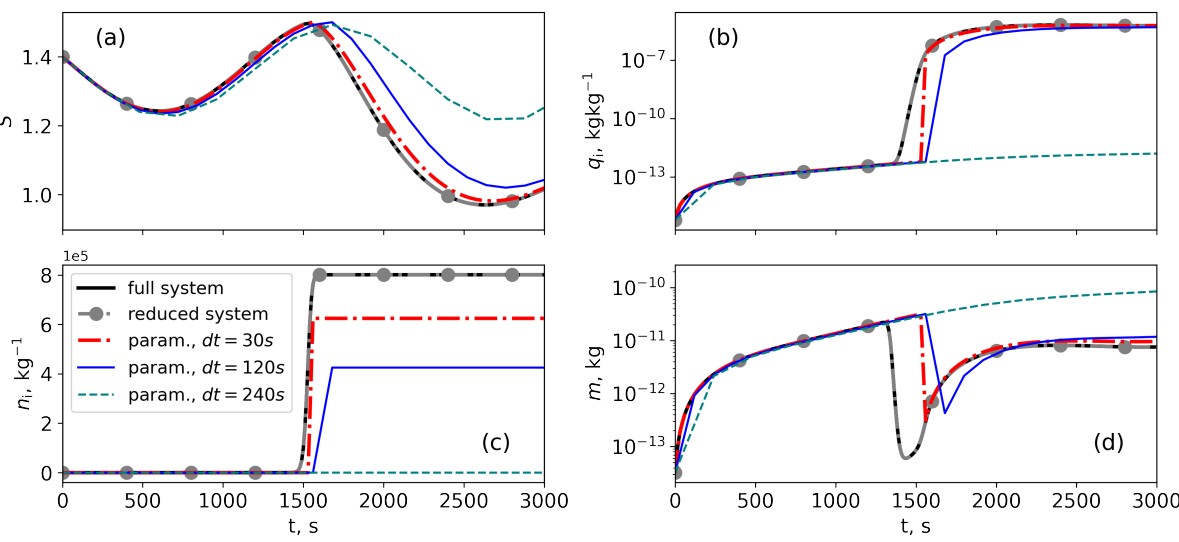

**Figure 14.** Comparison of the parameters evolution in full system, reduced system, and parameterisation with proposed correction. Initial conditions: $S = 1.4$, $n_\mathrm{i} = 0.018 \ \mathrm{kg}^{-1}$, $q_\mathrm{i} = 6 \cdot 10^{-16} \ \mathrm{kg\,kg}^{-1}$.

nucleation event, the difference in the final ice number concentration is increased to 46%. Other variables and the evolution of the slowly varying part are described well in cases where the nucleation is captured as such. Considering different cases and referring to the representation of the PDFs, a time step of 60 seconds can be suggested for parameterisation for the optimal


result. Further increase in time step leads to a decrease in accuracy, and in cases of the largest time step of 4 minutes, even to a wrong representation of the process, where some events are not captured. Such behaviour is observed due to the fact that the pre-existing ice grows too fast, depleting the available water vapor and thus preventing the saturation ratio from increasing to values triggering the nucleation.

## 4 Conclusions

This study aimed to extend the applicability of the nucleation parameterization Dolaptchiev et al. (2023) to a broader range of atmospheric conditions. Specifically, it addressed the regime of low pre-existing ice, where the rapid evolution of the ice mixing ratio limits the validity of the asymptotic approach used in earlier work. To overcome this limitation, a generalisation of the parameterisation derived in Dolaptchiev et al. (2023), including the effect of variable ice mean mass, is proposed in the article. Compared to the constant mean mass assumption, the proposed approach has the advantage of incorporating a physically grounded correction to the growth coefficient, reflecting the actual variation in ice particle growth under different conditions. The considered method allows for the inclusion of the mean mass variation, which leads to a better representation of the predicted ice number concentration distribution.

The new approach is validated against ensemble simulations based on the full double-moment scheme, showing a good agreement with the exact solution. The representation of individual nucleation events is done with reasonable accuracy, and the mean relative error with respect to the full system is less than 16% for the ice number concentration, which is similar to the usual accuracy of measurements. A vast number of cases are represented accurately, but the substantial differences remain for the extreme cases of the high forcing term values. The probability density function of the occurring $n_i$ determined from independent ensemble simulations captures the major statistical characteristics, such as the mean and the center of the PDF.

Because of the computational efficiency demand for NWPs and climate applications, the effect of uncertainties resulting from large ice microphysics time steps in the proposed extended parameterisation is assessed as well. The optimal representation of individual nucleation events combined with the sufficiently well captured statistics of $n_{post}$ occurrence is achieved with the $dt = 60$ seconds. This recommendation is comparable to the fast physics time step of about 4 minutes, typically used in NWP models. Evaluation of the numerical effectiveness shows that parameterisation is several times faster than the solution of the full system with a time step of 1 second.

Given the strong agreement of the proposed method with resolved physics, its robustness, and sufficient accuracy, the approach is now ready for further application in NWP models, such as ICON. The generalization of the method allows for direct coupling with forcings from dynamical parameterizations, thereby enhancing its integration into operational systems. For instance, the detailed representation of local subgrid-scale GWs dynamical fields impacting cirrus clouds can be incorporated using the Multi-Scale Gravity Wave parameterization (MS-GWaM) (Bölöni et al., 2021; Voelker et al., 2024) coupled to the described ice nucleation parameterisation. Additionally, this coupling approach can be extended to other relevant dynamical effects, e.g., turbulence.

*Code and data availability.* The air parcel model source code and Python scripts for processing and plotting the data are available on Zenodo at https://doi.org/10.5281/zenodo.14244119. Data set from ICON simulation used in the construction of the initial conditions for the parcel model, as well as all output data used for plots, can be found on Zenodo at https://doi.org/10.5281/zenodo.13819511.

## Appendix A

Coupling of the ice physics processes to the GW theory is done under the assumptions of mid-frequency and high-frequency GWs, where the GW time scale $T_w$ is tied to the Brunt-Vaisala frequency $N = \sqrt{\frac{g}{\theta}\frac{d\bar{\theta}}{dz}}$ by the following relation:

$$T_w = \frac{1}{\varepsilon^\beta N}, \quad \beta = 0, 1; \quad \varepsilon = 0.1,$$

where $\beta = 0$ characterizes the high-frequency and $\beta = 1$ the mid-frequency GW time scale, $\varepsilon$ is an introduced small parameter used for defining the distinguished limit in the asymptotic analysis. For instance, the characteristic time scales in the conditions of the troposphere, where $N = 10^{-2}$ s$^{-1}$, are $T_w = 100$ and $T_w = 1000$ seconds for the high and mid-frequency waves, respectively. The diffusional growth time scale is estimated from $T_d = \left(Dm_c^{1/3}\frac{p_{si,c}}{p_{00}q_{v,c}}T_{00}n_c\right)^{-1}$ and has a value of about $\sim 340$ seconds. Here, $n_c$ is the reference ice number concentration, $m_c$ is the reference mean mass of ice particles. Given the characteristic time scales and expanding the Exner pressure $\pi$ in terms of a small parameter $\varepsilon$ one can write the following expression for the derivative of $\pi$ (in non-dimensional form, see Dolaptchiev et al. (2023)):

$$\frac{D\pi}{Dt} = \varepsilon^{2+\alpha}\frac{D\pi'}{Dt} + \varepsilon w \frac{d\bar{\pi}}{dz_s}, \tag{A1}$$

where $\pi = \left(\frac{p}{p_{00}}\right)^{\frac{R}{c_p}}$, $\alpha = 0$ corresponds to the strong and $\alpha = 1$ to the weak stratification case (Achatz et al., 2017, 2023), $\bar{\pi}$ is the Exner pressure of the hydrostatically balanced reference atmosphere, $R$ is the gas constant for dry air, and $\pi'$ describes the fluctuations due to the wave field.

As shown in Dolaptchiev et al. (2023), applying the scaling analysis, one can neglect the first term on the right-hand side in Eq. (A1). In the case where the leading order vertical velocity is solely due to a single GW, $w$ in the second term of Eq. (A1) can be written as

$$w = |\hat{w}|^{(0)}\cos(\omega t + \phi).$$

For a superposition of $j$ gravity waves, the following expression applies:

$$w = \sum_j |\hat{w}_j|\cos(\omega_j t + \phi_j).$$

Here $\hat{w}_j$, $\omega_j$, $\phi_j$ are the vertical wind amplitude, frequency, and phase of GW number $j$.

Substituting the following expressions in the Eq.(6) and also taking into account constant background updraft $w_{00}$, the last term can be written in the following form:

$$F(t) = \frac{gL_i}{c_p R_v T^2}\left(\sum_j \hat{w}_j \cos(\omega_j t + \phi_j) + w_{00}\right), \tag{A2}$$

where $L_i$ is the latent heat of sublimation, $g$ is the acceleration due to gravity, $c_p$ is the heat capacity, $R_v$ is the gas constant for dry air, $T$ is the background reference temperature.

**Table A1.** Values or ranges for the physical quantities within the relevant ranges for the UTLS region, see also (Baumgartner and Spichtinger, 2019; Dolaptchiev et al., 2023).

| Parameter | Description | Value or range |
|-----------|-------------|----------------|
| $T_{00}$ | Reference background temperature | 210 K |
| $\rho_{00}$ | Reference background density | $0.5 \text{ kg m}^{-3}$ |
| $\rho_i$ | Ice particles density | $920 \text{ kg m}^{-3}$ to $929 \text{ kg m}^{-3}$ |
| $p_{00}$ | Reference background pressure | 300 hPa |
| $J$ | Parameter for the nucleation rate | $4.9 \cdot 10^4 \text{ kg}^{-1}\text{s}^{-1}$ |
| $B$ | Parameter for the nucleation rate | 337 |
| $S_c$ | Critical saturation ratio over ice | 1.5 |
| $D$ | Growth parameter | $4.3 \cdot 10^{-8} \text{ kg}^{2/3}\text{s}^{-1}\text{K}^{-1}$ |
| $L_i$ | Latent heat of sublimation | $2.8 \cdot 10^6 \text{ Jkg}^{-1}$ |
| $c_p$ | Specific heat of dry air | $1005 \text{ J kg}^{-1}\text{K}^{-1}$ |
| $R$ | Gas constant for dry air | $287 \text{ J kg}^{-1}\text{K}^{-1}$ |
| $R_v$ | Gas constant for water vapor | $416 \text{ J kg}^{-1}\text{K}^{-1}$ |
| $p_{si,c}$ | Reference saturation pressure over ice | 1 Pa |
| $n_c$ | Characteristic ice number concentration | $2 \cdot 10^6 \text{ kg}^{-1}$ |
| $m_c$ | Characteristic mean ice crystal mass | $10^{-12} \text{ kg}$ |
| $\hat{m}$ | Mean mass of newly nucleated ice particles | $10^{-16} \text{ kg}$ |

**Appendix B**

The parameter selection and limitations of the proposed fit are further evaluated using additional results obtained with a more advanced fit formulation for $m_{eff}$. This version incorporates a third parameter, $m^*$, and introduces a more complex functional dependency, expanding the simple fit to a third-order polynomial and increasing the number of fitting coefficients to 17. This extension is designed to better capture the spread observed in the dataset shown before in Fig. 4.

    The PDFs of the relative errors of the $m_{eff}$, mean radius of ice particle $R_0$, and $n_{post}$, predicted with the use of the more
complex fit, with respect to the reference are shown in Fig. B1. As mentioned in section 2.3, the improvement in the target parameter $n_{post}$ is minor. The PDFs of predicted $m_{eff}$ and $n_{post}$ (see Fig. B2) indicate a better representation of the maximum and right tail of the PDF of ice number concentration. Despite those improvements, the use of a more comprehensive fit is not considered further, due to efficiency reasons and divergent behavior of the polynomials in the extreme cases.

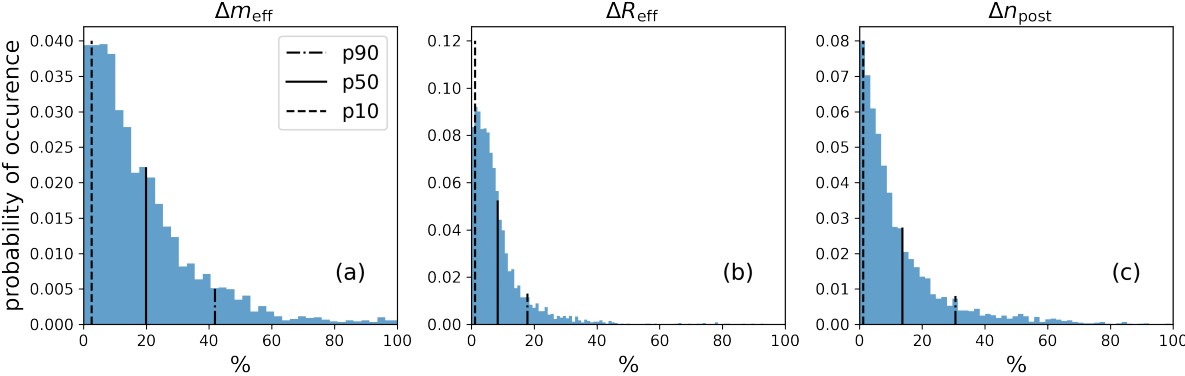

**Figure B1.** Probability density functions of the relative errors in the predictions of $m_{\text{eff}}$, $R_{\text{eff}}$, and $n_{\text{post}}$ made with the advanced fit, with respect to the exact solutions. Vertical lines correspond to the 90th, 50th, and 10th percentiles of the distribution.

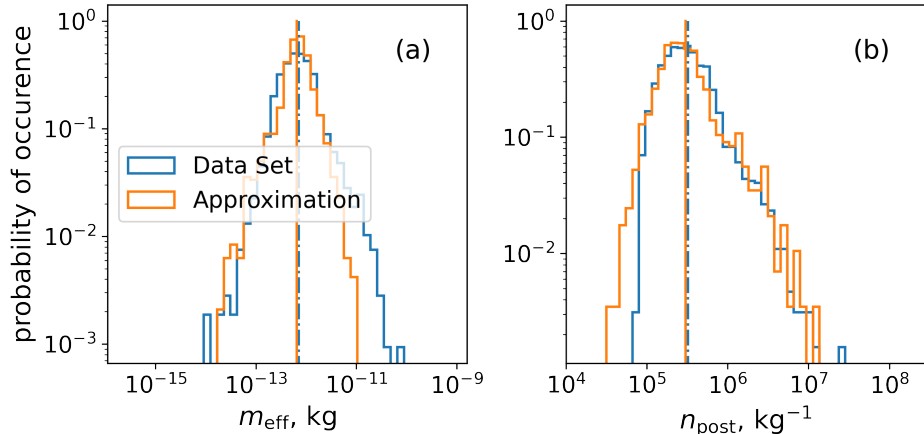

**Figure B2.** Probability density functions for $m_{\text{eff}}$, $n_{\text{post}}$, and $q_{\text{i}}$ exact ensemble solutions and predictions made with the advanced fit. Vertical lines correspond to the mean of the distribution.

## Appendix C

In order to further quantify the proposed parameterization and test it for the larger vertical velocity perturbations, a set of additional calculations was performed using the constant updraft and extreme amplitudes of GW-induced vertical velocity. The results are compared to the commonly used homogeneous nucleation parameterizations such as Kärcher and Lohmann (2002) and a reference system (1)-(3) with resolved ice physics. The ice number concentration prediction is presented depending on constant vertical updraft and $\hat{w}$ vertical updraft created by the GW (see Fig.C1). The proposed parameterization shows a good

agreement with the traditional approaches and captures the larger $n_{\text{i}}$ for the larger vertical updraft, which was also found by Kärcher and Lohmann (2002) using a bin model. Cases where vertical updraft was forced by the GWs also show the same

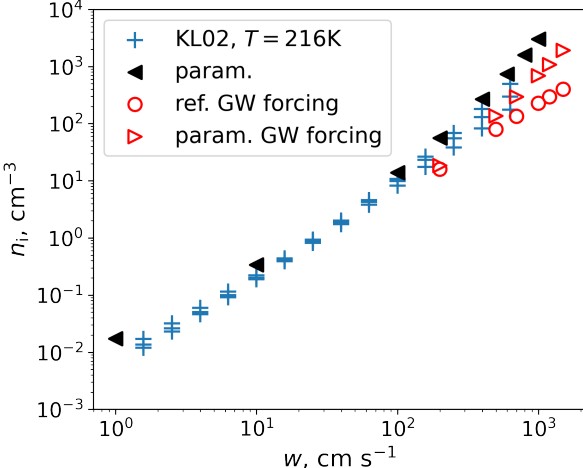

**Figure C1.** The ice number concentration calculated based on the proposed approach with the conditions of $T =$220K, time step of 10s and with the prescribed constant updraft (param.) and single GW forcing (param. GW forcing) is shown depending on the $w$ or $\hat{w}$. For comparisons, the results based on the reference system (1)-(3) without sedimentation are shown for the GW forcing (ref. GW forcing). As a reference for the constant updraft, the values from detailed microphysics calculations from Kärcher and Lohmann (2002) are shown, where the upper/middle/lower symbols correspond to an initialization of narrower/baseline/broader aerosol size distributions.

behavior as the constant updraft, however, the validation is done against the resolved version. The extended parameterization overestimates $n_i$ compared to the resolved physics, but the final predictions stay in the same order of magnitude and follow the expected physical trend. Notably, the larger deviations occur primarily in scenarios with extreme vertical velocity perturba-

tions, up to 15 ms$^{-1}$, which represent rare and highly energetic events in the UTLS region. Despite this, the parameterization successfully captures the qualitative and quantitative response of the system under such conditions. This demonstrates both the robustness and the flexibility of the approach, making it a reliable and efficient choice for representing a wide range of atmospheric scenarios, including those with strong dynamic forcing.

*Author contributions.* AK and SD proposed the concept of generalization of the parameterisation. The extension was built based on the

collaborative work of SD, PS, and UA, utilizing data produced from the ICON model with the gravity waves parameterisation developed within the group, under the leadership of UA. AK developed the methodology and conducted the investigation, defining the extended parameterisation with input from colleagues. The source code of the parcel model was developed by AK with the assistance of SD, scripts for data analysis and outputs were written and documented by AK. Data interpretation was carried out collectively by all authors. AK wrote the manuscript, incorporating contributions and revisions from all co-authors. SD, UA, and PS provided supervision and guidance throughout

the project.

*Competing interests.* The contact author has declared that none of the authors has any competing interests.

*Acknowledgements.* All authors thank the German Research Foundation (DFG) for support through the CRC 301 "TPChange" (Project-ID 428312742, Projects B06 "Impact of small-scale dynamics on UTLS transport and mixing", B07 "Impact of cirrus clouds on tropopause structure", and Z03 "Joint model development").

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
