# Peer review of "A new parameterisation for homogeneous ice nucleation driven by highly variable dynamical forcings"

_Geoscientific Model Development, 2024_

## Referee Comment (RC1)

**Review of "A new parameterisation for homogeneous ice nucleation driven by highly variable dynamical forcings?" by Kosareva et al. (gmd-2024-193)**

This study further develops a parameterization for estimating homogeneous ice nucleation for cirrus clouds forced by gravity waves. The work builds on a previous study by Dolaptchiev et al. (2023), which laid out the framework for predicting the number concentration of freshly nucleated ice crystals. Here, this parameterization is amended by a relationships that considers the initial ice mass, which is shown to depend on the initial ice concentration and the dynamical forcing. While the science presented here seems reasonable, its presentation requires significant revision. While most of my comments require rather minor requests for clarification, I cannot support the publication at the current stage.

**Major Comment**

Does the proposed parameterization of the initial ice mass (10) consider all relevant parameters? How do the authors know that the initial ice concentration and the dynamic forcing are the most relevant parameters? Considering that (10) causes a mean error of 30 % (ll. 191 – 192), there must be some parameters that influence the initial ice mass that are not considered in (10). From a simple adiabatic parcel perspective, I would assume that the linear size of the ice crystals should have an impact. I understand that the authors aimed to address this by considering $m_\ast$ (ll. 126 – 128, 193 – 198) – but with little success. Maybe, there is a better estimate for the size of the initial ice size? More comments on this issue are necessary.

**Minor Comments**

Ll. 1 – 2: For what models is the parameterization developed? Two-moment cloud microphysics in numerical weather or climate prediction models?

L. 3: What does the "correction" correct?

L. 4: The "double-moment ice microphysics" are used in the ensemble simulations or with the newly developed parameterization?

L. 10: Are the "system results" the ensemble results mentioned before?

Ll. 11 – 13: The extension to other dynamical forcings than gravity waves is not discussed in this study.

Ll. 25 – 31: Add a reference to Kärcher et al. (2024) on the impact of turbulence on homogeneous ice nucleation.

L. 38: Be specific about what the "deposition coefficient" refers to.

Ll. 39 – 40: To what "mass" does the "mean mass variability" refer to?

Ll. 45 – 46: To what does the "wide range of initial conditions" refer to? What is varied?

Ll. 58 – 59: While I agree that assuming a spherical shape simplifies the description of ice crystals, this comes with a lot of caveats. The authors should comment on this.

Ll. 56 – 58: Here, "$n_i$" and "$q_i$" are used to refer to the ice crystal concentration and the ice mixing ratio, respectively. In Eqns. (1) to (3), and other places, "$n$" and "$q$" are used. Are there differences between these variables? If not, why do the authors use a different notation?

Eqns. (1) and (2): The term containing "$J \exp(B(S-S_c))$" should not be identical in these equations. I believe it should be multiplied with "$n$" in Eqn. (2).

Eqn. (2): What is "$p$"?

L. 67: Based on Tab. A1, "$S_c$" seems to vary. Is this variability considered in the ensemble simulation? Or is a constant value assumed? What causes the variability?

Ll. 70 – 71: Eqn. (1) does not consider ice crystal growth due to deposition.

Ll. 74 – 76: I would not consider a parcel simulation as "realistic". Maybe, one could qualify them as "detailed".

Eqn. (5): Why is the ratio "$p/p_{si}$" opposite to the one used in Eqn. (2)?

L. 92: Multiple terms on the right-hand-side of the equation need to be defined.

L. 94: To what does the "prototype parameterisation" refer to?

Eqn. (7) ff.: Does this capital "$N$" refer to ice crystal concentrations such as the minor "$n$" used before? Why is a different notation introduced?

Eqn. (7): While I understand that this equation comes from Dolaptchiev et al. (2023), a few words on what is represents and how it is derived are necessary.

Ll. 101 – 109: While the first sentence indicates that this discussion refers to Fig. 1, I got lost in the subsequent sentences to what these statements refer to. Please clarify.

Fig. 1: What does the blue line indicate? What do the dash-dotted lines show?

Fig. 2: I believe the dashed gray line has the same meaning as the blue line in Fig. 1. Why are different colors used?

Ll. 126 – 128, 193 – 198: For what is "$m^\ast$" relevant? Later, it is stated that this quantity might enable a better parameterization. However, this quantity is not used any further as the improvement is minor. Why is this parameter introduced in the first place? As outlined in my major comment above, a few more details on the parameters chosen for Eqn. (10) are necessary, and they probably also justify the usage of "$m^\ast$".

Ll. 129 – 130: The authors should state earlier in the manuscript that solving the full system (1) to (3) is not the standard approach. The reduced system (4) to (6) is the commonly solved version.

Ll. 140 – 145: Define "ICON", "WKB", and "MS-GWab".

Fig. 4: Comment on the increase of $m_0$ with $n_{init}$, and the decrease of $m_0$ with $F(t_0)$. What are the physics behind this behavior?

L. 207: Is "initial $n$" identical to "$n_{init}$"? The corresponding statement should also be true for low $F(t_0)$. Should this be added?

Ll. 229 – 231: Why does the prediction of $q_{i,post}$ fail so much? I assume that $q_{i,post} = N_{post} * m_0$. However, $N_{post}$ and $m_0$ are predicted quite well. Thus, I guess there is a inherent co-variability between $N_{post}$ and $m_0$ that is not captured in the parameterization.

Ll. 245 – 246: State the equations to which the full system refers to. What is the "coupled system"? Is it identical to the reduced system (4) to (6)? Why does the "coupled system" not include Eqn. (4)?

Ll. 256 – 257: Do the "deviations in the forcing $F(t_0)$" refer to the change in $F(t_0)$ with time? Or is there an error in $F(t_0)$? Since $F(t_0)$ seems to be an external input, I cannot imagine the large errors that are presented here. Please clarify.

Ll. 280 – 281: Why is Eqn. (4) not considered?

Ll. 290 – 292: All distributions for $q_{i,post}$ seem to deviate quite substantially from the reference. Even for the shortest timestep.

**Technical Comments**

Ll. 16 ff.: The way other works are referenced is not in accordance with the Copernicus style guide.

L. 33: The "is" is not needed.

L. 39: Change "have" to "has".

L. 68: "Extner" to "Exner".

L. 91: Do not start this sentence with "Where".

L. 136: Change "gravity waves" to "GW".

L. 202: Change "bystability" to "by stability".

**References**

Dolaptchiev, S. I., Spichtinger, P., Baumgartner, M., & Achatz, U. (2023). Interactions between gravity waves and cirrus clouds: Asymptotic modeling of wave-induced ice nucleation. *Journal of the Atmospheric Sciences*, *80*(12), 2861-2879.

Kärcher, B., Hoffmann, F., Podglajen, A., Hertzog, A., Pluogonven, R., Atlas, R., ... & Gasparini, B. (2024). Effects of Turbulence on Upper-Tropospheric Ice Supersaturation. *Journal of the Atmospheric Sciences*, *81*(9), 1589-1604.

---

## Author Response (AR1)

We thank the reviewers for their constructive comments, which have helped improve the clarity and quality of our manuscript. Detailed responses to each comment are provided below. For clarity, changes, that are made in the manuscript are marked in blue, and the corresponding line numbers and/or sections from the marked-up manuscript are indicated.

**Anonymous Referee #1**

**Major Comment**

Does the proposed parameterization of the initial ice mass (10) consider all relevant parameters? How do the authors know that the initial ice concentration and the dynamic forcing are the most relevant parameters? Considering that (10) causes a mean error of 30 some parameters that influence the initial ice mass that are not considered in (10). From a simple adiabatic parcel perspective, I would assume that the linear size of the ice crystals should have an impact. I understand that the authors aimed to address this by considering $m_*$ (ll. 126 – 128, 193 – 198) – but with little success. Maybe, there is a better estimate for the size of the initial ice size? More comments on this issue are necessary.

**Response:** We appreciate the reviewer's careful analysis. We agree that this point requires further elaboration, and we have now extended our discussion accordingly. The major dependencies of the $m_0$ correction are connected directly to how it was defined from the full system calculation, via Eq. (9). There is a clear dependence on the parameters $F(t_0), n_{pre}, n_{post}$, however, one can only use the first two and correct for a possible variation in $n_{post}$ by considering the $m^*$. To illustrate the dependence, the ensemble calculations are presented in Fig. 4 in the manuscript with additional sets for fixed values for $n_{init}$ and $F(t_0)$. One can mention that there is a slight spread from the dependency, which happens for less than 5% of the cases. The addition of the third parameter $m^*$ was supposed to target this spread in the dataset. The results using the advanced fit with additional parameter $m^*$ are added below to the response (see Figs. 1 and 2), and to the manuscript in Appendix B.

As further support, we would like to mention that the PDF of the deviations now includes the mean radius of the ice particles, which is represented well, despite the effective mass deviations. In fact, the mean mass enters the growth equation with a power of 1/3, i.e., the radius plays a more important role compared to the mean mass. The results show that an error of 20% in mass translates into an error of less than 10% for the radius of ice crystals. Given the clear dependence of the data on two parameters and a good agreement of the target parameter $n_{post}$ as well as ice crystal radius, the authors would tend to support the use of a simple fit. We acknowledge that there might be a better solution to this problem, for instance, a data-driven approach. However, it lies outside of the scope of the current work, and we would like to stress that a simpler fit with physically based behavior is more reliable, especially in the NWP and climate applications.

**Changes:** The mean radius of the ice crystals is added to the analysis on lines 314-320 and Fig. 7:

The effective radius is calculated based on the effective mass as follows: $R_{eff} = \left( \frac{3 m_{eff}}{4 \pi \rho_i} \right)^{\frac{1}{3}}$. The probability density function (PDF) of the deviations is presented in Fig.7. Here the deviations are calculated as $\Delta M = (M_{predicted} - M_{exact})/M_{exact} \cdot 100\,\%$, with $M = m_{eff}, R_{eff}, n_{post}$. The mean errors for $m_{eff}$ and $n_{post}$ predictions are 24% and 16%, respectively. Despite the substantial deviation in effective mass, the prediction for the effective radius shows a good agreement and the mean error is only about 10%. More importantly, 90% of the error for the prediction of the ice number concentration lies below 37%. The majority of the individual cases are represented sufficiently well, whereas outliers and poorly captured events with a deviation in $n_{post}$ above 50% correspond to less than 10% of the distribution.

The results using the advanced fit with additional parameter $m^*$ are added in Appendix B.

**Minor Comments**

1. Ll. 1 – 2: For what models is the parameterization developed? Two-moment cloud microphysics in numerical weather or climate prediction models?

   **Response:** Thank you for raising this important point. The proposed parameterization is designed for the double-moment ice scheme within the ICOsahedral Non-hydrostatic (ICON) modeling framework developed by the DWD and MPI Hamburg. However, due to its generalized formulation and robustness for larger time steps, it can also be applied to other models and used in various applications and grid resolutions, including numerical weather prediction (NWP) and climate modeling.

[Figure]

Figure 1: Probability density functions for deviations of the $m_{eff}$, $R_{eff}$, $n_{post}$ predictions using the advanced fit. Vertical lines correspond to 90th, 50th, and 10th percentiles of the distribution.

[Figure]

Figure 2: Probability density functions for the $m_{eff}$, $n_{post}$ predictions using the advanced fit from the exact solutions. Vertical lines correspond to the mean of the distribution.

**Changes** on line 8: numerical weather prediction and climate models.

2. L. 3: What does the "correction" correct?

   **Response:** The correction is introduced to account for the mean mass of ice particles variability due to the deposition process.

   **Changes** on lines 3-4: ... derived correction to the parameterisation based on a large data set of parcel model simulations to account for the mean ice particle mass variations.

3. L. 4: The "double-moment ice microphysics" are used in the ensemble simulations or with the newly developed parameterization?

   **Response:** We appreciate the reviewer's comment and have revised the sentence. Here, we refer to the ensemble calculations, which are performed using both approaches: time-resolved double moment scheme (4)-(6) and the one using the proposed parametrisation with correction, solving Eqs. (5)-(6) and predicting ice number concentration via expression (8). The version with resolved ice physics processes is used as a reference for the validation of the parametrisation.

   **Changes** on line 5: time-resolved double-moment ice microphysics ...

4. L. 10: Are the "system results" the ensemble results mentioned before?

   **Response:** We appreciate the reviewer for pointing this out and have revised the abstract to improve its clarity. The authors refer to the same ensemble calculations using the time-resolved double-moment ice scheme.

   **Changes** on lines 10-11: the reference full double-moment system results.

5. Ll. 11 − 13: The extension to other dynamical forcings than gravity waves is not discussed in this study.

**Response:** The authors have now clarified this aspect in the revised manuscript. The possibility of extension of the vertical velocity perturbation to the turbulence and other sources is briefly mentioned in the conclusion, and now the following discussion point is added in Section 2.2.

**Changes** on line 15: ... such as large scale motions or even turbulence;
On lines 245-247: Because of the general model formulation, one can add the vertical velocity contributions to the forcing term based on other information provided by the model, for example, by the turbulence scheme.

6. Ll. 25 – 31: Add a reference to Kärcher et al. (2024) on the impact of turbulence on homogeneous ice nucleation.

**Response:** We appreciate the reviewer's suggestion and have added the following citation.

**Changes** on line 30.

7. L. 38: Be specific about what the "deposition coefficient" refers to.

**Response:** In the current part, the authors are referring directly to the cited article. According to Gierens et al. (2003), one can give the following explanation: the deposition coefficient is a parameter controlling the growth of ice crystals in the kinetic regime when crystal size is comparable to the mean free path of air molecules.

**Changes** throughout the text: the coefficient D for the accuracy is now renamed to the growth coefficient.

8. Ll. 39 – 40: To what "mass" does the "mean mass variability" refer to?

**Response:** Here we refer to the mean mass of existing ice particles; this particular mass is used in the growth equation within the resolved physics scheme.

9. Ll. 45 – 46: To what does the "wide range of initial conditions" refer to? What is varied?

**Response:** We are referring to the initial condition selection for the ensemble calculation further in the text in section 2.2. The initial ice number concentration, mean mass of the pre-existing ice particles, and the forcing terms are varied in the calculations. The range in which the parameters mentioned above are varied corresponds to their variability in the UTLS zone valid for cirrus conditions.

**Changes** on line 55: e.g., dynamical forcing and environmental conditions.

10. Ll. 58 – 59: While I agree that assuming a spherical shape simplifies the description of ice crystals, this comes with a lot of caveats. The authors should comment on this.

**Response:** Thank you for raising this important point. We understand the limitations of assuming a spherical shape for ice crystals and have now clarified this aspect in the revised manuscript. The newly nucleated particles, considered in the present work, produced by homogeneous nucleation, are initially small enough that assuming a spherical shape is a reasonable approximation. This assumption is also used in former investigations and parameterizations (see, e.g., Spichtinger and Gierens (2009); Kärcher and Lohmann (2002)). However, it becomes less accurate as ice particles grow through deposition and aggregation, which can lead to different aerodynamic, radiative, and microphysical properties, as well as influence cloud dynamics and precipitation formation (Pruppacher et al., 1998; Mishchenko et al., 1996; Baran, 2009). To address this, we have added the clarification in the manuscript.

**Changes** on lines 73-88: The mentioned assumption is reasonable for very small, newly nucleated ice particles formed through homogeneous nucleation. However, as ice particles grow through deposition, their shape becomes more complex, introducing some uncertainty in the calculations. For consistency, we maintain the same shape assumption in deposition calculations, acknowledging that this introduces a certain error. Although a more sophisticated shape approximation would be beneficial for studying ice growth dynamics, this does not affect the developed correction and goes beyond the topic of the current study; former nucleation parameterisations also assume spherical ice particles at their early stage of life (see, e.g., Kärcher and Lohmann, 2002; Ren and Mackenzie, 2005; Barahona and Nenes, 2008).

11. Ll. 56 – 58: Here, $n_i$ and $q_i$ are used to refer to the ice crystal concentration and the ice mixing ratio, respectively. In Eqns. (1) to (3), and other places, "n" and "q" are used. Are there differences between these variables? If not, why do the authors use a different notation?

    **Response:** We appreciate the reviewer's attention to details. The notation is corrected in the updated version using the $n_i, q_i$.

    Changes are applied to Eqs. 1-3, 4-6, and figures.

12. Eqns. (1) and (2): The term containing "J exp(B(S-$S_c$))" should not be identical in these equations. I believe it should be multiplied with "n" in Eqn. (2).

    **Response:** We appreciate the reviewer for pointing this out. Indeed the nucleation term has a different form in equations (1) and (2), the equation (2) term is now corrected to:
    J exp(B(S-$S_c$)) $\hat{m}_0$, where $\hat{m}_0 = 10^{-16}$kg is the mean mass of newly nucleated ice particles.

13. Eqn. (2): What is "p"?

    **Response:** The $p$ is pressure, the corresponding correction is added after equation (2) on line 87.

14. L. 67: Based on Tab. A1, "$S_c$" seems to vary. Is this variability considered in the ensemble simulation? Or is a constant value assumed? What causes the variability?

    **Response:** We thank the reviewer for the valuable comment. The general formulation of the critical saturation ratio over ice is dependent on temperature. The effect of the variable critical saturation ratio has been found insignificant (Spichtinger et al., 2023), therefore, the current work has considered a fixed value of $S_c = 1.5$. The following discussion is added to the definition of the critical saturation ratio.

    Changes in Table 1 and on lines 93-95: In general formulation, the critical value of the saturation ratio over ice is temperature dependent; in the current work, the value is kept constant while other studies (such as Spichtinger et al. (2023)) show an insignificant influence of $S_c$ on nucleation processes.

15. Ll. 70 – 71: Eqn. (1) does not consider ice crystal growth due to deposition.

    **Response:** We thank the reviewer for the comment. The text on lines 70-71 references the system of equations in general, where processes such as deposition and nucleation are taken into account. In the regime considered in the current work, the deposition does not contribute to the ice number concentration change, therefore, this term does not appear on the right-hand side of equation (1).

16. Ll. 74 – 76: I would not consider a parcel simulation as "realistic". Maybe, one could qualify them as "detailed".

    **Response:** The authors agree with the reviewer and replaced the description with 'detailed' on line 100.

17. Eqn. (5): Why is the ratio "$p/p_{si}$" opposite to the one used in Eqn. (2)?

    **Response:** We thank the reviewer for the careful reading. There was a typo in equation (5), which is now corrected in the manuscript.

18. L. 92: Multiple terms on the right-hand side of the equation need to be defined.

    **Response:** We thank the reviewer for the comment. The following definitions of the terms are added in the manuscript.

    Changes on lines 118-119: where $g$ is the acceleration due to gravity, $w_{00}$ is a constant background updraft, $\hat{w}_j$, $\Omega_j$, $\phi_j$ are the vertical wind amplitude, frequency, and phase of GW number $j$.

19. L. 94: To what does the "prototype parameterisation" refer to?

    **Response:** The prototype parametrisation refers to the equation (7) combined with the saturation ratio and ice mixing ratio diagnostic equations for the nucleation conditions.

    Changes on line 124: the prototype parameterization of the homogeneous nucleation process

20. Eqn. (7) ff.: Does this capital "N" refer to ice crystal concentrations such as the minor "n" used before? Why is a different notation introduced?

    **Response:** The capitalized $N$ was used following the paper Dolaptchiev et al. (2023), however, for better readability, we unified the parameters using the $n$.

    **Changes** throughout the text and in Eqs. 7,8,9.

21. Eqn. (7): While I understand that this equation comes from Dolaptchiev et al. (2023), a few words on what is represents and how it is derived are necessary.

    **Response:** The corresponding expanded discussion is added to the manuscript.

    **Changes** on lines 124-133: By employing asymptotic analysis, the authors systematically decompose the governing equations into different scale regimes corresponding to the key microphysical processes: nucleation and deposition. In the nucleation regime, where saturation reaches the critical value, they derive an analytical solution for the evolution of ice crystal number concentration. In the deposition regime, where ice crystals grow primarily by vapor deposition, the governing equations are solved separately under the assumption of a constant number of ice crystals. The solutions from both regimes are then matched using asymptotic matching techniques, ensuring a smooth transition between nucleation and subsequent ice growth. This approach results in a reduced analytical expression for the number of nucleated ice crystals per nucleation event, explicitly linking it to wave parameters (such as frequency and amplitude) and background atmospheric conditions (such as temperature and humidity).

22. Ll. 101 – 109: While the first sentence indicates that this discussion refers to Fig. 1, I got lost in the subsequent sentences to what these statements refer to. Please clarify.

    **Response:** The corresponding description is clarified in the corrected version of the manuscript.

    **Changes** on lines 144-155: One regime at high values of the pre-existing ice crystal number concentration (shown in Fig.1 in black) follows the same tendency as a constant mass case. The evolution of parameters can be separated into three phases (see Baumgartner and Spichtinger (2019)), where the nucleation part describes only the phase of rapid change of ice number concentration. In contrast, the saturation ratio and ice mixing ratio change slowly during this short time, and therefore can be assumed not to vary during this phase. However, conditions of low-to-zero pre-existing ice (shown in red) lead to a different behavior due to the growth coefficient change. Assumptions of the time separation for the nucleation regime do not hold, since the ice mixing ratio is changing on the same time scale as the ice number concentration. The extension to be proposed here aims to cover the whole range of initial conditions, allowing the use of the parameterisation accounting for mass variability independently from the regime.

23. Fig. 1: What does the black line indicate? What do the dash-dotted lines show?

    **Response:** The additional description is added to the caption of Fig. 1: The blue line corresponds to the critical saturation ratio, dashed lines mark the time of the nucleation event, where $S = S_c$ in red for the case $n_{init} = 10^2$ kg$^{-1}$, in black for the case $n_{init} = 10^6$ kg$^{-1}$.

24. Fig. 2: I believe the dashed gray line has the same meaning as the black line in Fig. 1. Why are different colors used?

    **Response:** We thank the reviewer for pointing out this inconsistency. The gray line also corresponds to the critical saturation ratio value, and the consistent (blue) color is now used throughout the figures.

25. Ll. 126 – 128, 193 – 198: For what is $m^*$ relevant? Later, it is stated that this quantity might enable better parameterization. However, this quantity is not used any further as the improvement is minor. Why is this parameter introduced in the first place? As outlined in my major comment above, a few more details on the parameters chosen for Eqn. (10) are necessary, and they probably also justify the usage of $m^*$.

    **Response:** The introduction of the $m^*$ is only used for the more complex dependency, which is due to the lack of improvement was removed from the results. To account for the major comment

as well as the present question, the authors are elaborating on the point of parameter choice in the corrected manuscript.

**Changes:** Highlight figures for the more complex fit are added to the Appendix B.

26. Ll. 129 – 130: The authors should state earlier in the manuscript that solving the full system (1) to (3) is not the standard approach. The reduced system (4) to (6) is the commonly solved version.

**Response:** The corresponding change is applied in the discussion of the system (4)-(6).

**Changes** on lines 119-121: It is worth mentioning that the application of the full system of Eqs. (1)-(3) is not a standard approach, whereas the reduced system (4)-(6) is close to the reduced model in Baumgartner and Spichtinger (2019), and variations of it are commonly used in NWP and climate applications, however including the ice mass concentration.

27. Ll. 140 – 145: Define "ICON", "WKB", and "MS-GWab".

**Response:** The corresponding definitions for ICON and WKB are added to lines 195 and 197, and the MS-GWaM definition is given on line 196.

28. Fig. 4: Comment on the increase of $m_0$ with $n_{init}$, and the decrease of $m_0$ with F($t_0$). What are the physics behind this behavior?

**Response:** We appreciate the reviewer's comment. To enhance clarity and ensure a more comprehensive discussion, we have revised the manuscript as follows.

**Changes** on lines 255-261: The increase of the mass with an increase of the initial number concentration is connected to the fact that pre-existing ice suppresses the nucleation process, hence the number of newly nucleated ice is smaller. Because the initial ice mean mass is quite high, and the new particles are 4 orders of magnitude lighter, the recalculated mean mass would increase due to the depositional growth if more pre-existing ice is present. The observed behavior of the mass in dependence on the forcing term is the opposite, it decreases with increasing $F(t_0)$. A larger forcing term at the point of the nucleation is associated with the larger updrafts, leading to a faster change in the saturation ratio over ice. The depletion of the water vapor via deposition in such conditions is ineffective, leading to smaller mass values.

29. L. 207: Is "initial n" identical to $n_{init}$? The corresponding statement should also be true for low F($t_0$).

**Response:** The following correction is made in order to have a uniform notation. The underestimation of the $m_0$ is observed for higher $n_{init}$ values and low values of $F(t_0)$, the corresponding adjustment is made in the text on line 303.

30. Ll. 229 – 231: Why does the prediction of $q_{i,post}$ fail so much? I assume that $q_{i,post} = N_{post} * m_0$. However, $N_{post}$ and $m_0$ are predicted quite well. Thus, I guess there is an inherent co- variability between $N_{post}$ and $m_0$ that is not captured in the parameterization.

**Response:** We appreciate the reviewer's comment. The reason for the large differences in $q_i$ prediction just after the nucleation is connected to the way a re-initialization is made, using the $m_0$. The explanation states that $m_0$ is not the mass after the nucleation event, which holds an inconsistency from the start. However, as shown in Figs. 13, 14 in the manuscript, such a prediction leads to a physically behaving system, and $q_i$ is approaching the reference solution within the next time step. For more details, see the discussion of the major comment above. In order to avoid confusion, we removed the comparison of the $q_i$ right after the nucleation event, since the most correct comparison follows the Figs. 13, 14 in the manuscript. The more fair approach would be to compare the values after several time steps after the nucleation.

**Changes** on lines 341-348: Apart from the values of $n_i$, the prediction for the ice mixing ratio $q_i$ is made based on the estimated mass $m_{eff}$ and the ice number concentration $n_{post}$, calculated using the corrected parameterisation, via $q_{i,post} = m_{eff}n_{post}$. Such an approach contains some physical inconsistencies because the approximated ice mean mass $m_{eff}$ differs from the mean mass after the nucleation event, which is supposed to be used. However, the estimation gives a suitable approximation for the further simulation of $q_i$. Further, the evolution.

31. Ll. 245 – 246: State the equations to which the full system refers to. What is the "coupled system"? Is it identical to the reduced system (4) to (6)? Why does the "coupled system" not include Eqn. (4)?

   **Response:** The full system references the system of Eqs. (1)-(3) without the sedimentation terms. Here coupled system refers to Eqs. (5) and (6) in particular, because the prediction of the ice number concentration is done by the parametrisation (7) or corrected parametrisation (8).

32. Ll. 256 – 257: Do the "deviations in the forcing $F(t_0)$" refer to the change in $F(t_0)$ with time? Or is there an error in $F(t_0)$? Since $F(t_0)$ seems to be an external input, I cannot imagine the large errors that are presented here. Please clarify.

   **Response:** The deviation in $F(t_0)$ the authors refer to is the error in the estimation of the value depending on the time step size, which stems from the $t_0$ approximation. In the current work forcing term $F(t)$ is calculated within the model with the initial conditions such as vertical updraft and GW parameters $\hat{w}, \omega, \phi$. The following clarifications are added to the text on line 370.

33. Ll. 280 – 281: Why is Eq. (4) not considered?

   **Response:** In this section, authors describe the sensitivity of the proposed approach using the extended parametrization to the time step size. The parametrised version requires one to calculate Eqs. (5) and (6) with the relatively large time step, and utilize Eq. (7) or (8) for the prediction of the ice number concentration. This is to be clarified in the text.

34. Ll. 290 – 292: All distributions for $q_{i,post}$ seem to deviate quite substantially from the reference. Even for the shortest timestep.

   **Response:** The poor estimation of the ice mixing ratio is purely associated with the assumed expression $q_{i,post} = m_0 n_{post}$, due to the fact, that $m_0$ does not correspond to the mean mass after the nucleation. The authors mention this limitation above in the answer to question 30.

35. Ll. 16 ff.: The way other works are referenced is not in accordance with the Copernicus style guide.

   **Response:** Thank you for pointing this out. We have carefully reviewed the references and updated them to comply with the Copernicus style guide.

36. L. 33: The "is" is not needed.

37. L. 39: Change "have" to "has".

38. L. 68: "Extner" to "Exner".

   **Response to 36-38:** We thank the reviewer for catching these errors. The following corrections are made in the manuscript on lines 38, 45, 90 and 476.

39. L. 91: Do not start this sentence with "Where".

   **Response:** Thank you for your suggestion. We have revised the sentence on line 116 to improve clarity.

40. L. 136: Change "gravity waves" to "GW".

41. L. 202: Change "bystability" to "by stability".

   **Response to 40, 41:** The following corrections are made in the manuscript on lines 187, 290.

**Anonymous Referee #2**
**General comments**

I / Legibility of the figures: please include labels to all figures (a, b...) so that they can be referenced in the caption and text. The very dense scatter plots in figures 5 and 6 are not very informative due to the extreme overlap of the symbols. Perhaps consider adding median and quantiles (as, e.g., in the scatter plots of Kramer et al., 2016, their figure 2). Figures 9, 10 and 11 would be clearer with an x-axis for the pdfs.

**Response:** We appreciate the reviewer's valuable suggestions regarding the clarity and legibility of the figures. We have carefully revised the figures and added the corresponding labels. For Figures 5 and 6, to improve clarity, we have incorporated additional statistical summaries, including median and quintiles, following the suggested approach from Krämer et al. (2016). Figures 9, 10, and 11 have been updated to include x-axes for the probability density functions (PDFs), enhancing readability. We believe these modifications significantly improve the visual presentation of our results. Thank you for these constructive suggestions.

Changes in Figs. 5,6 and 9-11.

II / To develop a parameterization based on a limited set of numerical simulations of nucleation, it is indeed important to assess the degree of realism of the initial conditions and forcing used to construct this dataset. The authors touch on that point but should elaborate further on it.

**Response:** We appreciate the reviewer's insightful comment on the importance of assessing the realism of the initial conditions and forcing. We have expanded our discussion in the revised manuscript, providing further justification for our chosen conditions, referencing relevant observational and modeling studies, and acknowledging potential limitations. We hope that these clarifications, included in Section 3, strengthen the validity of our approach.

Changes on lines 227-230: Moreover, the vertical velocity distributions in this range are comparable to those at higher altitudes, and the proposed parameterization is additionally tested under conditions representative of stronger forcing (see Appendix C). These tests suggest that the approach remains robust and physically consistent even beyond the original range of sampled data.

on lines 240-247: Further support for the realism of the applied conditions comes from Fig. 3 b, where the PDF of gravity wave–induced vertical velocity perturbations $\hat{w}$. The spread and frequency of occurrence show a good agreement with available observations from balloon measurements (Podglajen et al., 2016; Köhler et al., 2023).

1) Regarding the forcing, Figure 3 and the associated discussion should be made more quantitative and the choices should be explained. I suggest

    a) explaining the choices made to build the forcing dataset: why is the background updraft not always included? are tropical regions excluded ($f = 10^{-4} s^{-1}$) ?

    **Response:** We acknowledge this as an important aspect, and the arguments behind the chosen forcing need to be clarified. The selection of the forcing was dictated by the possible forcings observed in the ICON model with MS-GWaM GW parametrisation switched on. The background updraft is not always included also for the reason of mimicking the ICON cell-based quantities, where large-scale $w$ is close to zero in a substantial number of cells. As for the frequencies of the GWs, we considered a range from $f$ up to $N$ as specified on line 167 as well as in Fig. 3 (c), therefore tropical regions are included.

    Changes on lines 219-220: It is worth mentioning that the GW diagnosed from the MS-GWaM model dominates in cases 3 and 5 due to the larger amplitudes of the vertical velocity perturbations.

    b) extending the altitude range (currently 9 to 14 km as specified on line 165) to about 18 km in the tropics to include the climatically important TTL cirrus, which this parameterization should also represent.

    **Response:** We appreciate the reviewer's comment and agree that the TTL region above 14 km is relevant for the developed parameterization and nucleation processes. For the developed

[Figure]

Figure 3: Probability density functions for vertical velocity from ICON.

fit, we have reasons to believe that extending the altitude range and refitting the free parameters is not necessary in the current context. Our parameterization directly incorporates the local forcing conditions at the point of nucleation. As shown in Fig. 4 in the manuscript, a clear and consistent dependence on vertical forcing is observed. The correction we propose reflects the expected physical behavior under increasing forcing, and we, therefore, believe that the parameterization already captures the essential variability relevant for higher altitudes. Additionally, the range of vertical velocities perturbations observed between 8 and 18 km, shown in Fig.3 in the answer, is comparable to that in the 8–14 km range. Since our ensemble setup includes randomized background updrafts and superimposed gravity waves, the sampled conditions are representative of the broader UTLS, including the TTL. To further support this, we include results from test cases with high vertical velocities under both constant updraft and gravity wave-driven conditions (Figs. 4, 5 in the answer). These are compared with resolved physics and the predictions from Kärcher and Lohmann (2002), demonstrating that the corrected parameterization provides consistent and physically meaningful results even under stronger forcing conditions. The corresponding discussion and comparison to the Kärcher and Lohmann (2002) scheme are added to the Appendix C.

**Changes** on lines 225-227: While this range does not explicitly cover the full tropical tropopause layer (TTL), including altitudes up to 18 km, we note that the selected range captures the majority of ice formation conditions observed in the Upper Troposphere/Lower Stratosphere (UTLS).

c) including a normalization for the first three panels in Fig. 3 (with units). In particular, clarify what is shown in the 3rd panel and whether it relates to the power spectral density of $w$.

**Response:** We have revised Fig. 3 as well as the caption. Fig. 3 (c) corresponds to the intrinsic frequency of gravity wave outputs from the ICON-MS-GWaM run.

d) including more references to observational studies, for instance Podglajen et al. (2016) or Kohler et al. (2023). The first one is dedicated to observations of w which the authors are using to drive their simulations. The second contains comparisons with models, and some information on geographic variability. From a quick look (to be confirmed by the authors), it seems that the dataset of the authors underestimates vertical wind variability.

**Response:** Following the suggestion, we added the citations in the paper and attached the PDF of the vertical velocity perturbations from the MS-GWaM, which can be compared to the observations provided in Podglajen et al. (2016); Köhler et al. (2023). Referring to Fig.3, we would like to point out that, in our view, the variability in the dataset shows reasonable agreement with the observations.

**Changes** on lines 243-244: The spread and frequency of occurrence show a good agreement with available observations from balloon measurements (Podglajen et al., 2016; Köhler et al., 2023).

e) considering assessing other quantities than momentum fluxes, which do not directly affect ice microphysics. Note that the fact that a model can simulate quite well some GW-induced quantities (momentum fluxes) does not necessarily mean that others (vertical wind) are well

[Figure]

Figure 4: Evolution of the $n_i, q_i$ parameters under the conditions of large vertical velocity perturbations from the GW: $\hat{w} = 10$ ms$^{-1}$ (upper panel), $\hat{w} = 15$ ms$^{-1}$ (lower panel).

represented (see, e.g., Podglajen et al, 2020).

**Response:** We thank the reviewer for this comment. However, we would like to clarify that our study does not aim to evaluate the model's overall representation of gravity-wave-induced dynamics, since it was done in the previous studies (Bölöni et al., 2021). Instead, our focus is on selecting physically reasonable initial conditions for the simulations. Based on the vertical velocity PDF, the statistics appear to be captured to some extent, but a detailed quantitative comparison with observations is beyond the scope of this study. Additionally, the form of the proposed parameterization is directly dependent on the forcing, ensuring that the resulting response is physically meaningful. We have clarified this point in the revised manuscript.

**Changes** are reflected in Appendix C.

2) A more thorough description of the simulations' initial conditions is required. How exactly were $n_i$ and $m_i$ selected ? On lines 137-139, it is written "Other conditions are varied randomly using a uniform distribution in physically meaningful ranges motivated from the paper (Krämer et al. (2016)): we choose $10^{-4}kg^{-1} < n_{init}(t = 0) < 10^7 kg^{-1}$, and $10^{-16}kg < m_{mean}(t = 0) < 10^{-12}$kg". What is meant here by uniform distribution? An arguably wise choice would be a uniform distribution in $log(Ni)$ (to be consistent with Kramer et al.), but this sentence means a uniform distribution in $Ni$, i.e. oversampling large $log(Ni)$ (see, e.g., the distributions in Jensen et al., 2013). Also, the lower end of your Ni values would be smaller than $10^{-11}$ cm$^{-3}$, which is well below the range shown in Kramer et al. (2016). Please explain.

**Response:** We thank the reviewer for the valuable comment and agree that the description of the initial conditions was indeed not fully explained. The initial number concentration $n_i$ is, in fact, selected from a uniform distribution in $log(n_i)$, as suggested by Krämer et al. (2016). This is illustrated in Figure 4, where it is seen that large values of $n_i$ are not oversampled. We did not clearly state this in the manuscript, and we revised the description to clarify this point. Additionally, we included cases with no initial ice present, which explains the lower bound of $n_i$ values, corresponding to $10^{-11}$ cm$^{-3}$. We updated the manuscript accordingly to better explain these choices.

**Changes** on lines 189-193: Other initial conditions are varied randomly using a uniform distribution in logarithmic space, i.e., $\log(n_{init})$ in physically meaningful ranges motivated by observations (Krämer et al., 2016): we choose $10^{-4}$ kg$^{-1} < n_{init}(t = 0) < 10^7$ kg$^{-1}$, and $10^{-16}$ kg

[Figure]

Figure 5: The ice number concentration prediction based on the proposed parameterization depending on a vertical updraft. Initial conditions: $n(0) = 10^{-20}$ kg$^{-1}$, $T(0) = 220$K, vertical updraft varied from 0.01 to 10 ms$^{-1}$.

$< m_{mean}(t = 0) < 10^{-12}$ kg. The lower bound for the initial number concentration is chosen to represent the conditions of the low-to-no pre-existing ice, which can occur in applications to the global NWP models.

**Specific comments:**

1. l 16: for citations, here and elsewhere in this context, please use \citep instead of (\cite or \citet{})

   **Response:** We appreciate the reviewer's suggestion and have corrected the manuscript accordingly.

2. l 17: "meaning the" → "meaning THAT the"

   **Response:** The following correction is made in the manuscript on line 20.

3. l22-23: unclear, please clarify

4. l 26: "lower" than what ? should it be "low"

5. l 28: "for the description of " by "to describe"

6. l 33: "is extended" by "has extended"

7. l 60: "pressure" and temperatures

   **Response to 3-7:** We thank the reviewer for the careful reading. The following corrections and clarifications are made in the manuscript on lines 25-26, 31, 33, 38 and 80.

8. Equations 1-3: it would be helpful to clarify which parameters depend on $T$ ($S_c$ ? $J$ ?)

   **Response:** We appreciate the reviewer's suggestion and added the comment on the following topic in the manuscript. The critical saturation ratio over ice in general is temperature dependent; however, in the current work, it was considered as a constant value of $S_c = 1.5$. We also added a discussion of the choice $S_c$ in section 2.1. The nucleation rate $J$, on the other hand, is constant and defined in Table 1.

   **Changes** on lines 93-95: In general formulation, the critical value of the saturation ratio over ice is temperature dependent; in the current work, the value is kept constant while other studies (such as Spichtinger et al. (2023)) show an insignificant influence of $S_c$ on nucleation processes.

9. n , q are not defined, but $n_i$, $q_i$ are introduced above. The homogeneity of all terms in equation 2 should be checked.

   **Response:** We thank the reviewer for the careful reading. The notation is unified using $n_i$, $q_i$.

   **Changes** are applied to Eqs. 1-3, 4-6, and figures.

10. l 68:"Extner" → "Exner"

11. l 75: "valid during long periods of the nucleation" → "valid over the duration of the nucleation"

   **Response to 10, 11:** The following corrections are made in the manuscript on lines 90, 476 and 102.

12. l91: for clarity, you could define F(t) as a function of the Exner function tendency

   **Response:** The authors thank the reviewer for the comment. Details on the coupling of gravity waves to the scheme, as well as the definition of the forcing term, are provided in Appendix A. In the corrected manuscript, we would prefer to retain the time-dependent formulation of the forcing, as it is directly relevant to the subsequent discussion.

13. lines 95 and 119: couldn't you directly include the crystal's mass parameter $m_0$ instead of these D* and D** which I didn't find elsewhere in the manuscript

   **Response:** We appreciate the reviewer's suggestion and have revised the notations accordingly.

   **Changes:** D* and D** are removed from the text.

14. l 129: m0 appears here for the first time, but has not been defined before (the reader can guess it is q0/n0)

   **Response:** For better readability the correction $m_0$ is now introduced within the equation (9) and has been replaced by the effective mass $m_{eff}$, to avoid further confusion.

   **Changes:** the effective mass is now introduced in Eq. 8.

15. l 142-144: "This parameterisation is based on WKB theory (Bölöni et al. (2021)) and implemented using Lagrangian ray volumes, which are considered as carriers of the GW fields' wave-action density.": consider moving the reference to Bölöni et al. (2021) at the end of the sentence - here it could be interpreted as WKB theory is due to Bölöni et al.

   **Response:** The following correction is made in the manuscript on lines 196-197.

16. l 169: the chosen reference is not the best suited here; perhaps use VanZandt (1982) or one of the older references which proposed this form of the GW spectrum for the atmosphere

   **Response:** We appreciate the reviewer for pointing this out. The following correction has been made on line 236.

17. l 173: it seems to me that this may be an understatement and the vertical wind is actually underestimated compared to available observations

   **Response:** The following statement has been removed.

18. l 193-199: the extension is not shown- does it need to be mentioned?

   **Response:** The following description has been expanded due to Reviewer 1's comments, and an additional figure with the results for the improved fit is incorporated in the Appendix B.

19. l 207: I find it difficult to assess the agreement with this figure.

   **Response:** We thank the reviewer for the comment. The authors tried to improve the clarity of Fig. 6 by adding the median and quantiles of the distributions.

20. l 261: missing "are"

21. l 273: higher → larger

22. l 343: consider referring to Karcher et al. (2024) about turbulence and cirrus

   **Response to 20-22:** We appreciate the reviewer's suggestions and have corrected the manuscript accordingly. The corrections are made on lines 370, 382, 30.

23. Figure 9: clarify the caption. Please explain why the number of horizontal bars varies.

    **Response:** We thank the reviewer for the comment. The clarity of the caption has been improved by adding the description and additional text in the discussion of the figure.

    Changes on lines 364-366: In Figs. 9-11, horizontal bars represent percentiles of the distribution to illustrate data variability. Note that in several cases, the 10th percentile lies near zero and may not be easily distinguishable in the plots due to scale limitations.

24. In the bibliography section, some references are missing the doi/url

    **Response:** We have carefully reviewed the references and updated them in accordance with the comment.

**References**

Barahona, D. and Nenes, A.: Parameterization of cirrus cloud formation in large-scale models: Homogeneous nucleation, Journal of Geophysical Research-Atmospheres, 113, https://doi.org/10.1029/2007JD009355, 2008.

Baran, A. J.: A review of the light scattering properties of cirrus, Journal of Quantitative Spectroscopy and Radiative Transfer, 110, 1239–1260, https://doi.org/https://doi.org/10.1016/j.jqsrt.2009.02.026, xI Conference on Electromagnetic and Light Scattering by Non-Spherical Particles: 2008, 2009.

Baumgartner, M. and Spichtinger, P.: Homogeneous nucleation from an asymptotic point of view, Theoretical and Computational Fluid Dynamics, 33, 83–106, https://doi.org/10.1007/s00162-019-00484-0, 2019.

Bölöni, G., Kim, Y.-H., Borchert, S., and Achatz, U.: Toward transient subgrid-scale gravity wave representation in atmospheric models. Part I: Propagation model including nondissipative wave–mean-flow interactions, Journal of the Atmospheric Sciences, 78, 1317–1338, https://doi.org/10.1175/JAS-D-20-0065.1, 2021.

Dolaptchiev, S. I., Spichtinger, P., Baumgartner, M., and Achatz, U.: Interactions between Gravity Waves and Cirrus Clouds: Asymptotic Modeling of Wave-Induced Ice Nucleation, Journal of the Atmospheric Sciences, 80, 2861 – 2879, https://doi.org/https://doi.org/10.1175/JAS-D-22-0234.1, 2023.

Gierens, K. M., Monier, M., and Gayet, J.-F.: The deposition coefficient and its role for cirrus clouds, Journal of Geophysical Research: Atmospheres, 108, https://doi.org/10.1029/2001JD001558, 2003.

Köhler, L., Green, B., and Stephan, C. C.: Comparing loon superpressure balloon observations of gravity waves in the tropics with global storm-resolving models, Journal of Geophysical Research: Atmospheres, 128, e2023JD038 549, https://doi.org/10.1029/2023JD038549, 2023.

Krämer, M., Rolf, C., Luebke, A., Afchine, A., Spelten, N., Costa, A., Meyer, J., Zoeger, M., Smith, J., Herman, R. L., et al.: A microphysics guide to cirrus clouds–Part 1: Cirrus types, Atmospheric Chemistry and Physics, 16, 3463–3483, https://doi.org/10.5194/acp-16-3463-2016, 2016.

Kärcher, B. and Lohmann, U.: A parameterization of cirrus cloud formation: Homogeneous freezing of supercooled aerosols, Journal of Geophysical Research: Atmospheres, 107, AAC 4–1–AAC 4–10, https://doi.org/https://doi.org/10.1029/2001JD000470, 2002.

Mishchenko, M. I., Travis, L. D., and Mackowski, D. W.: T-matrix computations of light scattering by nonspherical particles: A review, Journal of Quantitative Spectroscopy and Radiative Transfer, 55, 535–575, https://doi.org/https://doi.org/10.1016/0022-4073(96)00002-7, light Scattering by Non-Spherical Particles, 1996.

Podglajen, A., Hertzog, A., Plougonven, R., and Legras, B.: Lagrangian temperature and vertical velocity fluctuations due to gravity waves in the lower stratosphere, Geophysical Research Letters, 43, 3543–3553, https://doi.org/10.1002/2016GL068148, 2016.

Pruppacher, H. R., Klett, J. D., and Wang, P. K.: Microphysics of clouds and precipitation, 1998.

Ren, C. and Mackenzie, A. R.: Cirrus parametrization and the role of ice nuclei, Quarterly Journal of the Royal Meteorological Society, 131, 1585–1605, https://doi.org/10.1256/qj.04.126, 2005.

Spichtinger, P. and Gierens, K. M.: Modelling of cirrus clouds–Part 1a: Model description and validation, Atmospheric Chemistry and Physics, 9, 685–706, https://doi.org/10.5194/acp-9-685-2009, 2009.

Spichtinger, P., Marschalik, P., and Baumgartner, M.: Impact of formulations of the homogeneous nucleation rate on ice nucleation events in cirrus, Atmospheric Chemistry and Physics, 23, 2035–2060, https://doi.org/10.5194/acp-23-2035-2023, 2023.